# Decoupled electrolysis for hydrogen production and hydrazine oxidation via high-capacity and stable pre-protonated vanadium hexacyanoferrate

Fei Lv[1], Jiazhe Wu[1], Xuan Liu[1], Zhihao Zheng[1], Lixia Pan[1], Xuewen Zheng[1], Liejin Guo [1] & Yubin Chen [1] ✉

Decoupled electrolysis for hydrogen production with the aid of a redox mediator enables two half-reactions operating at different rates, time, and spaces, which offers great flexibility in operation. Herein, a pre-protonated vanadium hexacyanoferrate (p-VHCF) redox mediator is synthesized. It offers a high reversible specific capacity up to 128 mAh g$^{-1}$ and long cycling performance of 6000 cycles with capacity retention about 100% at a current density of 10 A g$^{-1}$ due to the enhanced hydrogen bonding network. By using this mediator, a membrane-free water electrolytic cell is built to achieve decoupled hydrogen and oxygen production. More importantly, a decoupled electrolysis system for hydrogen production and hydrazine oxidation is constructed, which realizes not only separate hydrogen generation but electricity generation through the p-VHCF-$N_2H_4$ liquid battery. Therefore, this work enables the flexible energy conversion and storage with hydrogen production driven by solar cell at day-time and electricity output at night-time.

Hydrogen ($H_2$) is considered one of the most promising alternatives to traditional fossil fuels due to its zero carbon emissions and high energy density (120 MJ kg$^{-1}$)[1–3]. Among various methods for hydrogen production, water electrolysis is a sustainable and environmentally friendly technology that has been receiving lots of attention[4–7]. Conventional one step water electrolysis with diaphragm or membrane as the separator typically faces several critical challenges[8,9]. Firstly, using membranes will increase the system costs and restrict the direct use of the fluctuating renewable energy. Secondly, water electrolysis rate is limited by the sluggish kinetics of the oxygen evolution reaction (OER) because the hydrogen evolution reaction (HER) and OER are tightly coupled and the reaction kinetics of two half-reactions are interdependent to each other. Thirdly, the pressure differences between the sides of the separator and reactive oxygen species caused by the coexistence of $H_2$, $O_2$, and catalysts will accelerate the degradation of membrane, thus increasing safety issues[10–12].

To address this situation, Cronin et al. proposed a new electrolysis architecture, in which a soluble redox mediator of phosphomolybdic acid ($H_3PMo_{12}O_{40}$) was employed as electron-coupled proton buffer to decouple the one-step water splitting process into two steps[13]. This "decoupled" water electrolysis strategy with the aid of a mediator electrode enables the production of $H_2$ and $O_2$ at different rates, time, and spaces, which greatly increases the flexibility to harness the intermittent renewable energy. Compared with the soluble mediator, decoupled electrolysis system with solid-state redox mediators can avoid using membranes, which shows considerable prospects[14]. Recently, several solid-state redox mediators have been developed for decoupled water electrolysis systems[14,15]. For instance, our previous work presented a sodium nickelhexacyanoferrate mediator to decouple acid water electrolysis and amphoteric water electrolysis. The decoupled device was driven by one single perovskite solar cell with a solar-to-hydrogen efficiency of 14.4%[16]. Ma et al. introduced a Prussian

---

[1]International Research Center for Renewable Energy, State Key Laboratory of Multiphase Flow in Power Engineering, Xi'an Jiaotong University, Xi'an, Shaanxi 710049, China. ✉e-mail: ybchen@mail.xjtu.edu.cn

blue analogs of $Cu[Fe(CN)_6]_{2/3} \cdot 3.4H_2O$ (CuFe TBA) as a solid-state redox mediator to decouple the HER and OER in acid water electrolysis. The CuFe TBA electrode shows high rate performance and good cycling performance due to the Grotthuss proton conduction[17]. However, both the capacity and cycle performance of currently developed redox mediators cannot meet the high energy/power requirement and the research is still in the preliminary stage, which greatly limits the practical application of decoupled systems[18].

Considering the sluggish kinetics of OER, the anodic OER can be replaced by other anodic oxidation reactions that are kinetically and thermodynamically favorable for energy-saving $H_2$ production, such as the methanol oxidation reaction[19–21], the formate oxidation reaction[22,23], and the isopropanol oxidation reaction[24–26]. Among available options, hydrazine, as a liquid proton carrier, can be oxidized to inert nitrogen and release protons at the low voltage, effectively decreasing the input electrical energy[27,28]. Besides, hydrazine oxidation reaction (HzOR) can be combined with a reduction reaction with more positive potential to form a hydrazine battery/cell for simultaneous generation of electricity[29–31]. Inspired by this principle, combining hydrazine oxidation with the reduction of mediator electrode into a decoupled electrolysis system can enable hydrogen production and electricity generation, which may offer the possibility for the flexible energy conversion and storage using renewables. However, this topic in decoupled system is seldom investigated up to present.

Herein, we develop a pre-protonated vanadium hexacyanoferrate (p-VHCF) Prussian blue analog as a solid-state redox mediator with enhanced hydrogen bonding network. The p-VHCF electrode offers high-rate performance and long cycling stability, which can be used for efficient decoupled hydrogen production in acid water. Further, a decoupled electrolysis system for hydrogen production and hydrazine oxidation is constructed via the redox cycling of p-VHCF. This system not only separates $H_2$ generation at high-rate, but realizes the oxidation of hydrazine with electricity generation through the p-VHCF-$N_2H_4$

liquid battery. In such a manner, the flexible energy conversion and storage using renewables can be achieved.

## Results and discussion
### Structure and morphology of p-VHCF mediator
The prototype of the decoupled water electrolysis architecture is shown in Fig. 1a, which consists of the hydrogen evolution electrode, oxygen evolution electrode, and p-VHCF mediator electrode in a one-chamber cell. The operation of the decoupled cell involves two electrolysis processes. Specifically, the $H_2$ production step (Step 1) includes the oxidation of p-VHCF (p-VHCF$_{red}$ → p-VHCF$_{ox}$) in the anode and the reduction of $H^+$ in the cathode. The subsequent $O_2$ production step (Step 2) involves the anodic oxidation of $H_2O$ and cathodic reduction of p-VHCF (p-VHCF$_{ox}$ → p-VHCF$_{red}$). Step 1 and Step 2 can be cycled owing to the high reversible stability of p-VHCF. The pre-protonated p-VHCF with expanded hydrogen bond network displays fast Grotthuss proton conduction. Therefore, the HER and OER are successfully decoupled by the reversible redox reaction of p-VHCF in different time without introducing membrane separation. Alternatively, the OER process can be replaced by the hydrazine oxidation reaction. As shown in Fig. 1b, the p-VHCF$_{ox}$ formed during hydrogen production is coupled with the hydrazine oxidation process to generate a new electrolysis architecture to recycle the mediator. Considering the more positive potential of the reduction of p-VHCF than the hydrazine oxidation reaction (potential results are discussed below), the p-VHCF-$N_2H_4$ liquid battery can be formed, which enables hydrazine oxidation with simultaneous electricity generation (Step 2').

In this study, the vanadium hexacyanoferrate (VHCF) powder was synthesized via a simple wet-chemistry method, where the chemical reduction of $V_2O_5$ was first applied to form a vanadium precursor followed by the co-precipitation process. The pre-protonated VHCF (p-VHCF) was then obtained by chemical reduction of VHCF in a certain environment (see the experimental section)[32,33]. The crystal structure of p-VHCF is illustrated in Fig. 2a, where the lattice water in the

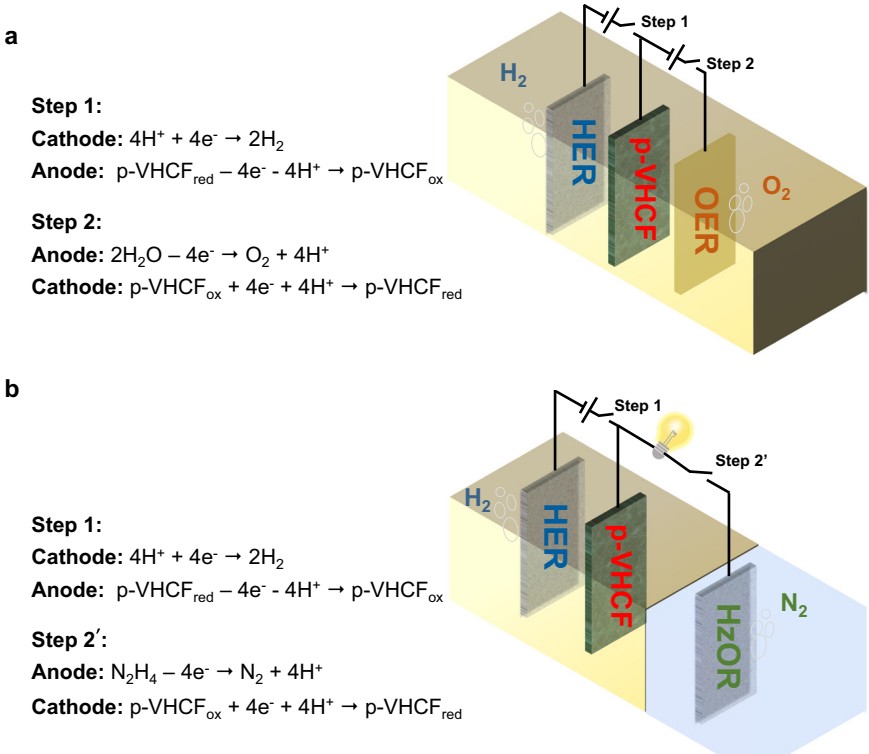

**a**

**Step 1:**
**Cathode:** $4H^+ + 4e^- \rightarrow 2H_2$
**Anode:** p-VHCF$_{red}$ $- 4e^- - 4H^+ \rightarrow$ p-VHCF$_{ox}$

**Step 2:**
**Anode:** $2H_2O - 4e^- \rightarrow O_2 + 4H^+$
**Cathode:** p-VHCF$_{ox}$ $+ 4e^- + 4H^+ \rightarrow$ p-VHCF$_{red}$

**b**

**Step 1:**
**Cathode:** $4H^+ + 4e^- \rightarrow 2H_2$
**Anode:** p-VHCF$_{red}$ $- 4e^- - 4H^+ \rightarrow$ p-VHCF$_{ox}$

**Step 2':**
**Anode:** $N_2H_4 - 4e^- \rightarrow N_2 + 4H^+$
**Cathode:** p-VHCF$_{ox}$ $+ 4e^- + 4H^+ \rightarrow$ p-VHCF$_{red}$

**Fig. 1 | Illustration of two-step decoupled electrolysis process. a** Schematic of the hydrogen/oxygen production from decoupled acid water electrolysis with p-VHCF mediator electrode. **b** Schematic of decoupled electrolysis for hydrogen production and hydrazine oxidation with p-VHCF mediator electrode.

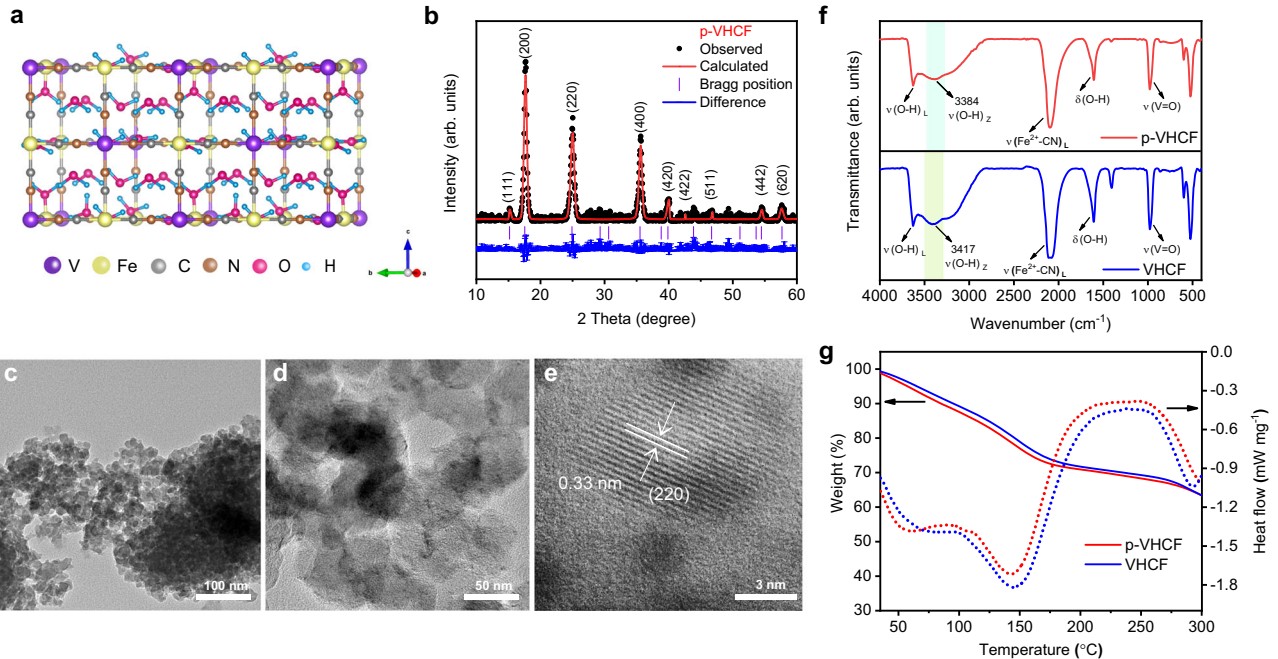

**Fig. 2 | Structure and morphology characterization. a** The structural illustration of p-VHCF crystal. **b** XRD pattern and Rietveld refinement profile of p-VHCF. **c**–**e** TEM images of p-VHCF at different magnifications. **f** FTIR spectra. **g** TGA and DSC curves of p-VHCF and VHCF.

framework can form hydrogen bond network to promote the transport of protons. To clarify the difference of crystal structures after the pre-protonated process, the X-ray diffraction (XRD) measurement was employed and the corresponding Rietveld refinement profiles are shown in Fig. 2b and Supplementary Fig. 1. The results show that VHCF and p-VHCF are well-assigned to a typical Prussian blue analog chemical compound of $Cu[Fe(CN)_6]_{2/3}$ (PDF No. 86-0513) with a face-centered cubic structure. The Rietveld refinement XRD profiles indicate that the lattice parameter of 10.145 Å ($R_{wp}$ = 7.91%, $\chi^2$ = 2.37) for p-VHCF is larger than 10.031 Å ($R_{wp}$ = 9.70%, $\chi^2$ = 2.99) for VHCF, which can be contributed to the introduced lattice water after the protonation process[17,34]. The transmission electron microscope (TEM) images show that the p-VHCF presents irregular particles with the diameter of 20–50 nm. The high-resolution TEM images indicate that the p-VHCF particles show good crystallinity and clear lattice fringes with a spacing of about 0.33 nm (Fig. 2c-e). The energy dispersive spectroscopy (EDS) element mapping of VHCF and p-VHCF also proved the homogeneous distribution of K, V, Fe, C, and N element (Supplementary Figs. 2 and 3).

Fourier transform infrared (FTIR) analysis was employed to determine the molecular structure after pre-protonation. As shown in Fig. 2f, the obvious FTIR peaks at 2097 and 979 cm$^{-1}$ for VHCF and p-VHCF correspond to the C≡N and V=O stretching modes. After being pre-protonated, the vibrational hydrogen bond O-H of VHCF shifts 33 cm$^{-1}$ from 3417 cm$^{-1}$ to 3384 cm$^{-1}$ (p-VHCF). This shift upon pre-protonation can be attributed to the enlargement of the hydrogen bonding network[35,36]. The thermogravimetric analysis (TGA) and differential scanning calorimetry (DSC) curves were used to determine the water content of the electrodes (Fig. 2g). The DSC curves exhibit apparent endothermic peaks at around 150 °C, which correspond to the loss of crystal lattice water molecules. The weight loss of 26.8% below 170 °C for p-VHCF is 1.48% higher than that of VHCF, indicating the higher content of crystal lattice water in p-VHCF. The coordination water can not only stabilize the crystal structure but also improve the osmotic hydrogen-bond network and facilitate proton conduction through the Grotthuss mechanism[17,32,34]. The composition of as-synthesized samples were analyzed by inductively coupled plasma emission spectrometry (ICP-OES) and the molecular formulae of p-VCHF and VHCF were determined as $K_{0.1}VO_{0.9}[Fe(CN)_6]_{0.8}\cdot4.8H_2O$

and $K_{0.2}VO_{0.5}[Fe(CN)_6]_{0.7}\cdot4H_2O$, respectively. Overall, these results proved that the protonation process of p-VHCF introduces crystal lattice water to expand the hydrogen bond network and improve the proton conduction, which is favorable to improve its electrochemical performance.

## Electrochemical properties and redox centers of p-VHCF mediator

The electrochemical properties of the p-VHCF electrode in acid electrolyte (6 M $H_2SO_4$) were then investigated through a three-electrode configuration with the Pt plate as counter electrode, Ag/AgCl as reference electrode, and p-VHCF electrode as working electrode. Figure 3a (black line) shows the cyclic voltammogram (CV) curve of p-VHCF obtained at a scan rate of 5 mV s$^{-1}$. There are three pairs of distinct redox peaks of p-VHCF, same as the VHCF electrode (Supplementary Fig. 4), which is attributed to the multistep reversible faradaic reactions. The potentials of HER and OER were also measured by linear sweep voltammetry (LSV) test using the commercial Pt-coated Ti-mesh electrode and $RuO_2/IrO_2$-coated Ti-mesh electrode, respectively. Clearly, the electrochemical window of p-VHCF lies between the onset potentials of the HER and OER, indicating that p-VHCF can be used as a redox mediator to decouple the acid water electrolysis. To further inspect the kinetics behavior of p-VHCF, CV measurement was recorded at different scan rates from 1 to 10 mV s$^{-1}$ (Supplementary Fig. 5). The peak current ($i_p$) and the scan rate ($v$) of the CV curves follow a power-law relationship ($i_p=av^b$, where a and b are constants). As shown in Fig. 3b, most calculated b values for the anodic peaks and cathodic peaks exceed 0.8 (0.89, 0.98, 0.79 for anodic $O_1$, $O_2$, $O_3$; 0.86, 0.98, 0.84 for cathodic $R_1$, $R_2$, $R_3$), which signifies an ultra-fast proton insertion/de-insertion kinetics[17,37,38].

Rate performance at various current densities was also evaluated. As shown in Fig. 3c, a reversible capacity of p-VHCF electrode was 152 mAh g$^{-1}$ at a current density of 2 A g$^{-1}$. Even at a high current density of 200 A g$^{-1}$, a respectable capacity of 47 mAh g$^{-1}$ was still achieved, indicating ultrafast proton insertion/de-insertion rate. When the current density was up to 300 A g$^{-1}$, the electrode only delivers a capacity of 17 mAh g$^{-1}$, indicating the limit of this electrode. The VHCF electrode shows lower rate performance compared with the p-VHCF electrode

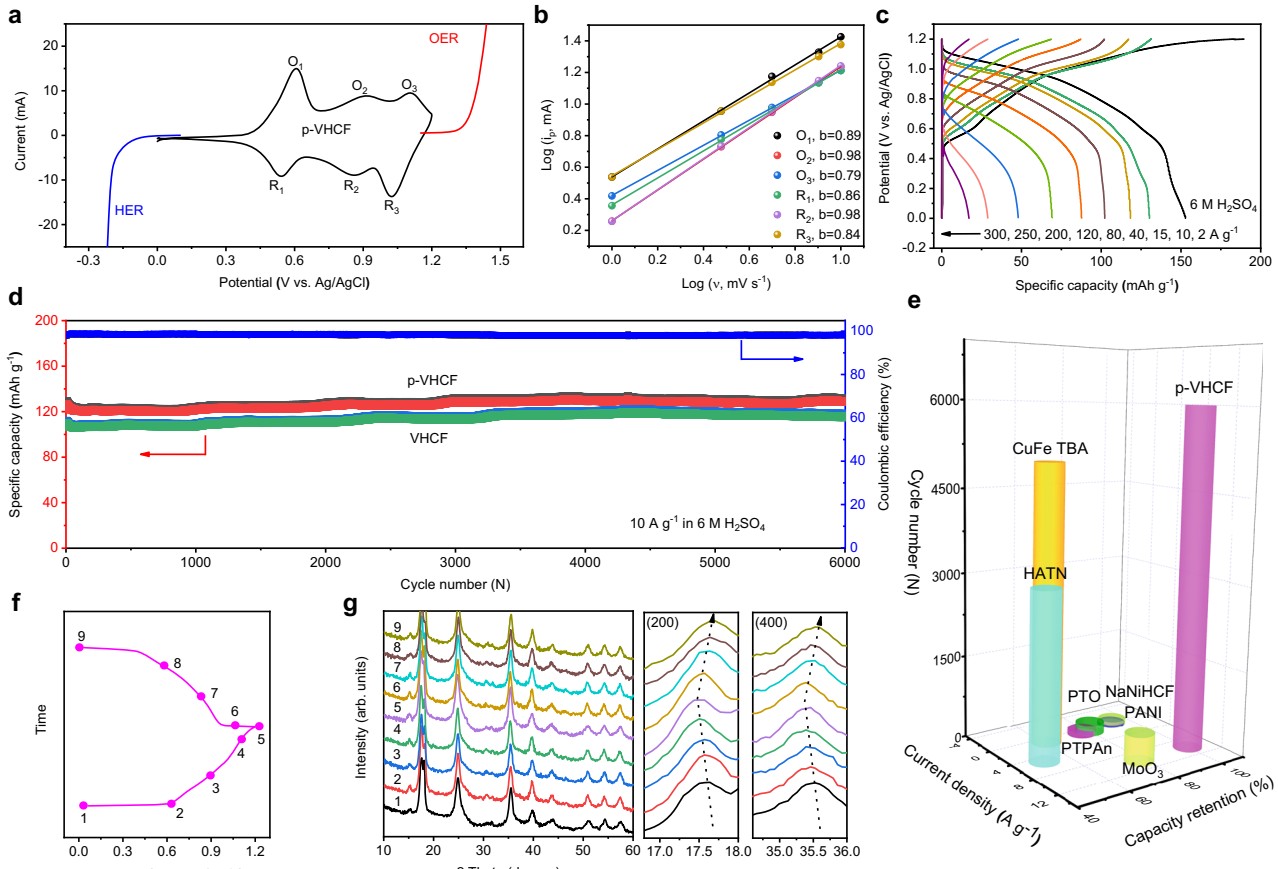

**Fig. 3 | Electrochemical performance of the mediator electrode. a** CV curve of the p-VHCF electrode (black line), the LSV curves of the Pt-coated Ti-mesh electrode for the HER (blue line) and the commercial RuO$_2$/IrO$_2$-coated Ti-mesh electrode for the OER (red line) at a scan rate of 5 mV s$^{-1}$ in 6 M H$_2$SO$_4$. **b** Relationship between the peak current (i$_p$) and scan rate (ν) of p-VHCF electrode. **c** Rate performance of the p-VHCF electrode for selected current densities. **d** Cycling performance at 10 A g$^{-1}$ for 6000 cycles. **e** Cycle performance of the p-VHCF redox mediator electrode compared to previously reported mediator electrodes in decoupled acid water electrolysis. **f** Charge/discharge potential profiles of the p-VHCF electrode (Points 2-8 correspond to the potentials of the three pairs of redox peaks in the CV curve). **g** Corresponding ex-situ XRD patterns at different charging/discharging states and the enlarged views of the (200) and (400) peaks in the XRD patterns.

(Supplementary Fig. 6). To evaluate the proton storage/release capacity of the electrodes, galvanostatic charge-discharge tests were employed with the potential window of 0–1.2 V (vs. Ag/AgCl). p-VHCF and VHCF electrodes both exhibit good capacity stability under long-term cycling, with about 100% retention of the initial capacity after 6000 cycles at 10 A g$^{-1}$ (Fig. 3d). In comparison, the p-VHCF electrode shows higher specific capacity up to 128 mAh g$^{-1}$. The XRD patterns, scanning electron microscope (SEM) images, and X-ray Photoelectron Spectroscopy (XPS) results after 6000 cycles were collected (Supplementary Fig. 7). There are not apparent changes of the structure and morphology after the long-term test, indicating the good stability of the p-VHCF electrode. As displayed in Fig. 3e, the p-VHCF mediator electrode depicts better cycling stability in terms of capacity retention and cycle number than most previously reported redox mediator electrodes for acid water electrolysis (Supplementary Table S1)[16,17,37–41].

We further tested the cycle performance of the p-VHCF electrode in various concentrations of H$_2$SO$_4$, including 0.5, 1, 3, 5, and 6 M, at 10 A g$^{-1}$. As shown in Supplementary Fig. 8, the p-VHCF electrode delivers a specific capacity of 108 mAh g$^{-1}$ and 90% capacity retention at 3 M concentration after 6000 cycles. It also has a specific capacity of 95 mAh g$^{-1}$ and can maintain more than 75% capacity retention at 10 A g$^{-1}$ after 6000 cycles as the H$_2$SO$_4$ concentration decreases to 0.5 M. The rate performance of p-VHCF electrode at varied current densities in 0.5 M H$_2$SO$_4$ electrolyte was exhibited in Supplementary Fig. 9, and those values remain favorable among the most currently

reported mediator electrode[8,9]. Generally, the properties of the current mediator are improved with the increased H$_2$SO$_4$ concentration. For the practical application, the reaction condition should be comprehensively considered in terms of the balance of the cost and performance. Higher electrolyte concentration can be acceptable, and a good example is commercial alkaline water electrolysis, where 30 wt.% (around 6 M) KOH aqueous solution is used as the electrolyte[42,43]. In addition, we tested the performances of the p-VHCF electrode in 10 M and 2 M H$_3$PO$_4$ electrolytes. As shown in Supplementary Fig. 10a, this material shows good cycling stability and a redox capacity of 98.8 mAh g$^{-1}$ at 10 A g$^{-1}$ in 10 M H$_3$PO$_4$. It is reported that the freezing point of H$_3$PO$_4$ electrolyte can be lower than −80 °C with the increase of electrolyte concentration to 9 M. This result indicates that the p-VHCF mediator has a potential application at ultralow temperatures for H$_2$ production[34,44]. Moreover, this electrode also shows considerable stability in 2 M H$_3$PO$_4$ electrolyte (Supplementary Fig. 10b). Owing to the good performances in different conditions, the p-VHCF electrode seems promising in decoupled electrolysis, batteries, and other electrochemical applications.

To better understand the effect of proton insertion/de-insertion on the structure change of the p-VHCF mediator, a series of in-situ/ex-situ measurements were carried out[32,33,45]. The ex-situ XRD patterns of the p-VHCF electrode at different charge/discharge states are presented in Fig. 3f, g. Clearly, there is no new peak appearing or original peak disappearing, indicating a non-phase transition process. While

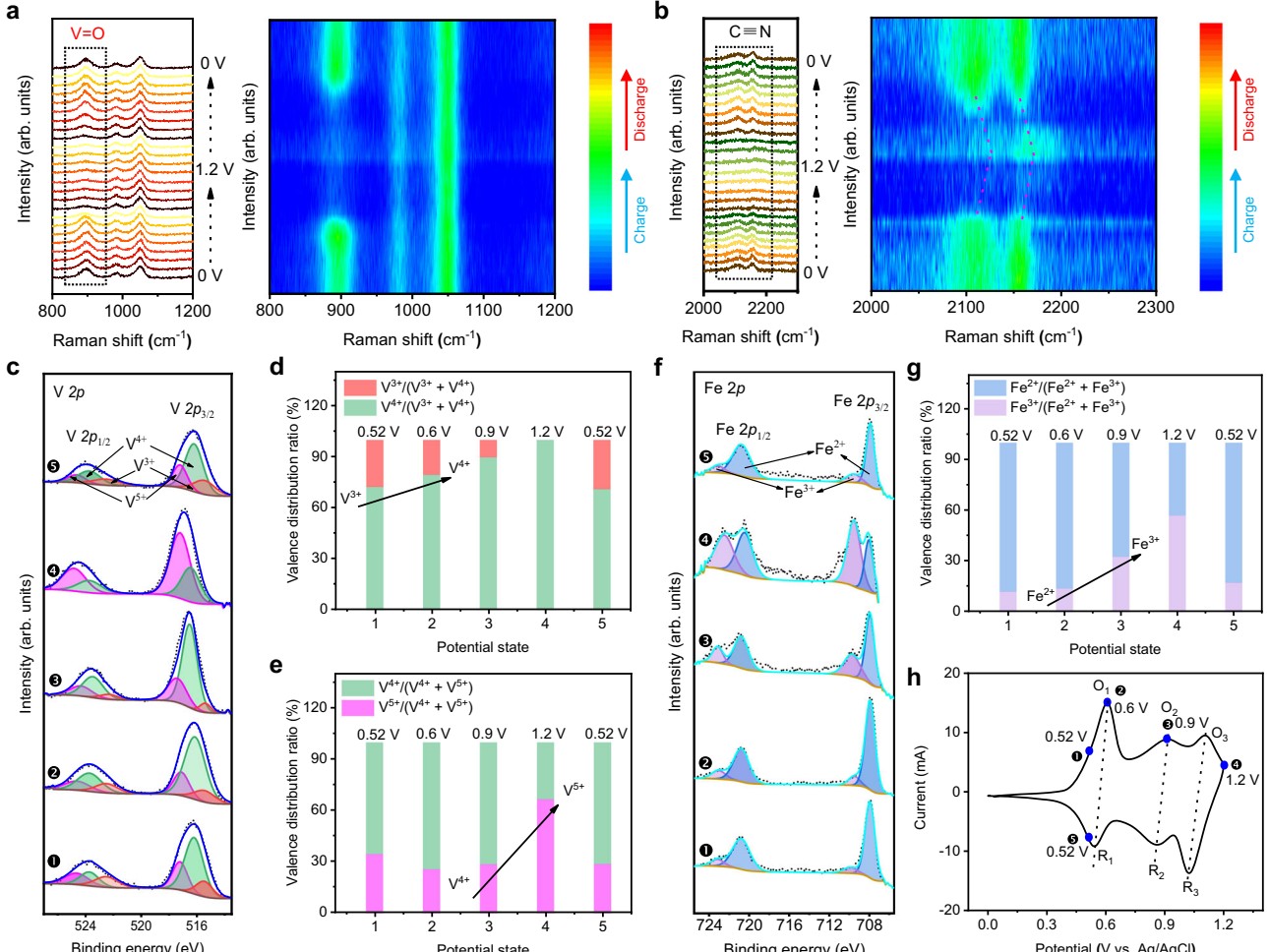

**Fig. 4 | In-situ/Ex-situ structure and composition measurement. a, b** In-situ Raman line spectra and 2D contour plots of the p-VHCF electrode. **c–g** Ex-situ XPS spectra of the p-VHCF electrode at different charging and discharging states (symbols 1-5, respectively, represent the states of 0.52, 0.6, 0.9, 1.2, and 0.52 V). **c** V 2p spectra and **d–e** valence distribution ratio diagrams of V. **f** Fe 2p spectra and (**g**) valence distribution ratio diagram of Fe. **h** CV curve of the p-VHCF electrode during the redox reaction.

during the charging process with the extraction of protons (from point 1 to 5 in Fig. 3f), the (200) and (400) diffraction peaks shift to lower degrees, implying an increase of the lattice parameter. Likewise, upon the discharge process (from point 5 to 9 in Fig. 3f), these diffraction peaks shift back to original states, revealing high reversibility of the electrode structure. This reversible lattice shrinkage and expansion during the proton insertion and de-insertion is attributed to the smaller radius of $[Fe(CN)_6]^{4-}$ than that of $[Fe(CN)_6]^{3-}$ anions[17,46].

In addition, the in-situ Raman and ex-situ XPS measurements were performed to illustrate the redox centers. As shown in the in-situ Raman spectra (Fig. 4a, b), the peak of V = O group located at 895 cm$^{-1}$ becomes gradually weakened during the charging process, while the peak intensity slowly increases during the discharging process. The peak position and intensity of C ≡ N groups bonded to iron ions located at 2110 and 2157 cm$^{-1}$ also change reversibly. Typically, the peaks slightly shift toward higher wavenumber and then reverses backward during the charging and discharging cycle, indicating the redox of $Fe^{2+}$ and $Fe^{3+}$ ions[47–49]. These results confirm that the V = O and $[Fe(CN)_6]^{4-}$ are the active redox sites during the redox process.

Ex-situ XPS measurement was further carried out to track the chemical status of components at various charging/discharging states (Fig. 4c-h). The partial V undergoes a shift of $V^{3+} \rightarrow V^{4+} \rightarrow V^{5+}$ during charging process from initial 0.52 V to 1.2 V (vs. Ag/AgCl) and reserves back during the discharging process. This further indicates that V ions

take part in the electrochemical redox reaction (Fig. 4c). According to the valence distribution diagram of V ions, the corresponding ratios of $V^{4+}/(V^{3+} + V^{4+})$ and $V^{5+}/(V^{4+} + V^{5+})$ are apparently increased at 0.6 V and 1.2 V, implying that the oxidation of $V^{3+}$ to $V^{4+}$ starts at the $O_1$ peak in the CV curve, and the oxidation of $V^{4+}$ to $V^{5+}$ starts at the $O_3$ peak in the CV curve (Fig. 4d, e). Meanwhile, the valance distribution ratio of $Fe^{3+}/(Fe^{2+}+ Fe^{3+})$ is significantly increased at 0.9 V during the charging process, indicating that the oxidation of $Fe^{2+}$ to $Fe^{3+}$ starts at the $O_2$ peak in the CV curve (Fig. 4f, g). Therefore, the three redox peaks of $O_1/R_1$, $O_2/R_2$, $O_3/R_3$ in the CV curve of the p-VHCF electrode (Fig. 4h) should respectively correspond to the redox reactions of $V^{3+}/V^{4+}$, $Fe^{2+}/Fe^{3+}$, and $V^{4+}/V^{5+}$, consistent with those results reported in the literature[33,46].

## Decoupled acid water electrolysis

Subsequently, a membrane-free decoupled acid water electrolyzer was constructed with a Pt-coated Ti mesh electrode (1.5 × 2 cm$^2$), a RuO$_2$/IrO$_2$-coated Ti mesh electrode (1.5 × 2 cm$^2$), and a p-VHCF electrode (1.5 × 2 cm$^2$ with the mass loading of p-VHCF of 60 mg cm$^{-2}$) to illustrate the hypothesis shown in Fig. 1a. The image of the electrolyzer is shown in Supplementary Fig. 11. Chronopotentiometry measurement was carried out to evaluate the performance of this decoupled system in 6 M H$_2$SO$_4$ under different applied currents (5-200 mA). Figure 5a shows the chronopotentiometry data (voltage and potential vs time) at

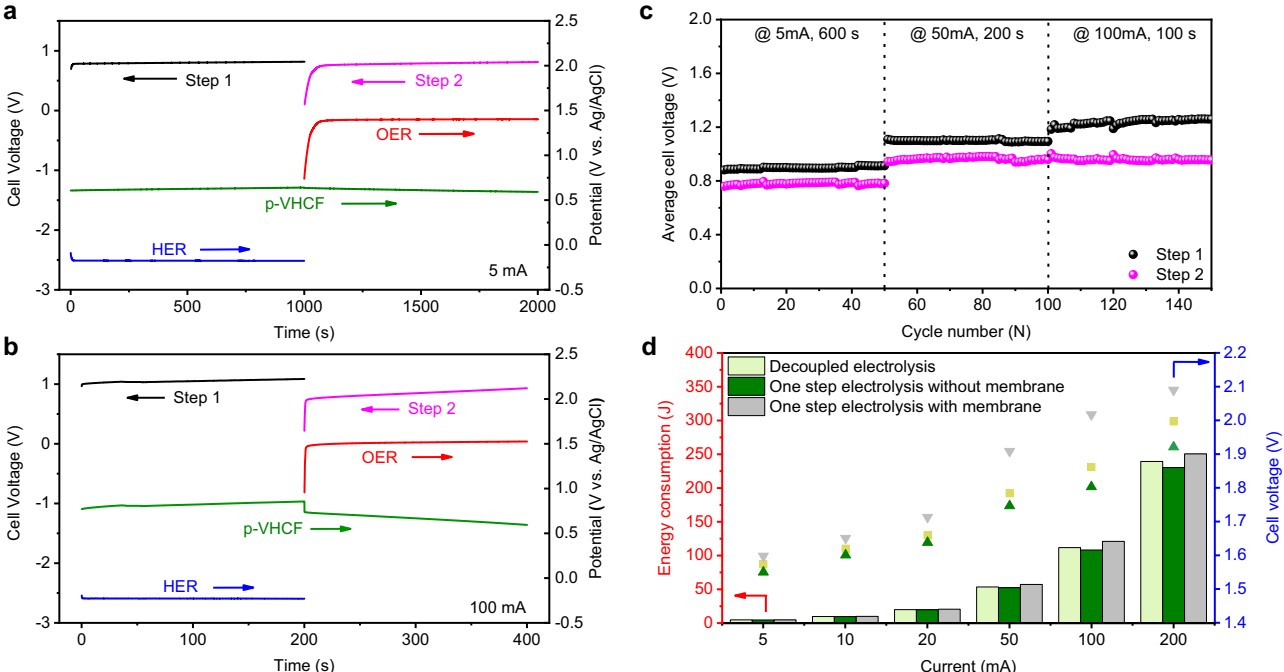

**Fig. 5 | Performance of the decoupled acid water electrolysis with the p-VHCF mediator electrode.** Chronopotentiometry curve (cell voltage vs time) of the cell at a current of (**a**) 5 mA and (**b**) 100 mA. The cell voltage of $H_2$ production (Step 1) and $O_2$ production (Step 2) are labeled by the black and magenta lines, respectively. The potentials of the OER electrode (red line), the HER electrode (blue line), and the p-VHCF electrode (cyan line) are also provided. **c** The stability performance of the separate $H_2/O_2$ generation at a current of 5 mA with a step time of 600 s, 50 mA with a step time of 200 s, and 100 mA with a step time of 100 s. **d** The energy consumption of the three systems (decoupled system, as well as one step water electrolysis with and without a membrane) at different currents during 600 s test.

the current of 5 mA with step time of 1000 s. The Step 1 (HER process) exhibits an average cell voltage of ~0.80 V, which comes from the potential difference between the anodic potential of p-VHCF oxidation of 0.63 V (vs. Ag/AgCl), and the cathodic potential of $H_2O$ reduction of −0.17 V (vs. Ag/AgCl). The average voltage of Step 2 (OER process) is 0.78 V, which is obtained from the anodic potential of $H_2O$ oxidation (1.39 V vs. Ag/AgCl) and the cathodic potential of p-VHCF reduction (0.61 V vs. Ag/AgCl). Therefore, a total voltage of 1.58 V was obtained for decoupled acid water electrolysis. This value is slightly higher than that of direct membrane-free water electrolysis with $RuO_2/IrO_2$-Pt electrodes (1.54 V) due to the internal polarization of the p-VHCF electrode. The two-step acid water electrolysis was further investigated at a high current of 100 mA with a step time of 200 s, showing that the cell voltages of steps 1 and 2 are 1.0 V and 0.84 V, respectively (Fig. 5b). The corresponding separate $H_2$ and $O_2$ generation can be verified by the corresponding videos given in Supplementary Movie 1. To reflect the operational flexibility of the decoupled system, the decoupled device was also operated at different currents from 10 to 200 mA (Supplementary Fig. 12). The efficiency of the decoupled system is calculated to be 97.8% at 100 mA compared to the corresponding one-step system according to the previous reports (Supplementary Fig. 13)[13,50]. The potential of the p-VHCF electrode as a function of time was acquired at different currents, as shown in Supplementary Fig. 14. The polarization gradually increases with the increase of applied currents from 5 to 100 mA, with the coulombic efficiency of 98% even at a high current of 100 mA.

Noticed that the electrolytic step time can be easily managed by varying the loading amount of redox mediator. When the mass loading of the p-VHCF was increased from 60 mg cm⁻² to 466 mg cm⁻², the electrolytic cell can be cycled with a step-time of 12 h, indicating the good performance of the mediator material over long-term charge/discharge processes (Supplementary Fig. 15). According to the gas chromatography (GC) data in Supplementary Fig. 16, the pure $H_2$ and $O_2$ can be detected in Step 1 and Step 2 process. The Faraday

efficiencies for $H_2$ and $O_2$ production were determined to be 99.2% at 20 mA (Supplementary Fig. 17). Meanwhile, hydrogen evolution as a function of varied current was obtained. As shown in Supplementary Fig. 18, with all step time of 800 s, the hydrogen production amount was increased with the increased current. Typically, this decoupled system achieves Faraday efficiencies of higher than 99% for hydrogen evolution at various currents. The cycle performance of this decoupled water electrolysis process was further investigated with applied currents of 10 mA, 50 mA and 100 mA (Fig. 5c). After 150 consecutive cycles, the cell voltage of Step 1 and Step 2 have no obvious change, indicating the good stability for the decoupled hydrogen and oxygen production. To assess the practicality of the system, the energy consumption of the decoupled system at different currents is compared with the conventional one step water electrolysis with and without a membrane (Fig. 5d and Supplementary Fig. 19). Compared with the one-step electrolysis with a membrane, the decoupled configuration is more energy efficient, showing a considerable potential for practical application.

## Decoupled electrolysis for $H_2$ production and $N_2H_4$ oxidation

For decoupled electrochemistry, the oxidized mediator needs to be recycled for sustained hydrogen production. The standard recycle process contains electrochemical reduction coupled with OER, which requires additional energy input due to the sluggish kinetics[24,50,51]. Alternatively, we solve this issue by adopting hydrazine oxidation reaction (HzOR) as the anodic half-reaction for removing hydrazine from acid waste water (Fig. 6a). The operation of the device involves a step for $H_2$ production (Step 1) and a step for hydrazine oxidation (Step 2'), where Steps 1 and 2' can be cycled like a rechargeable system. Considering the more positive potential of the reduction of p-VHCF than the hydrazine oxidation reaction, the simultaneous electricity is generated in Step 2'. The according reactions are shown in Fig. 1b, and the overall reaction is hydrazine electrolysis in the acidic environment.

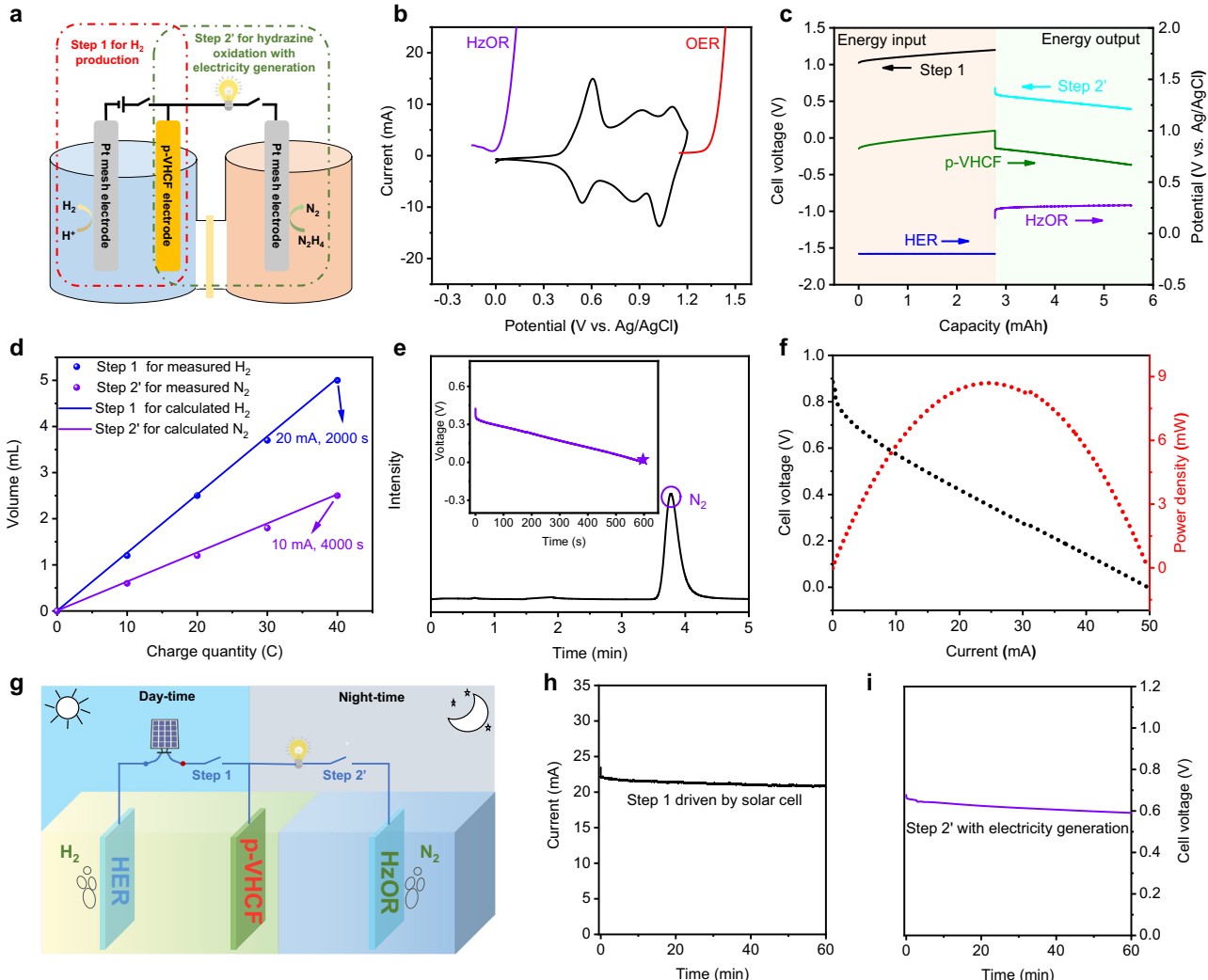

**Fig. 6 | Performance of decoupled electrolysis for H₂ production and hydrazine oxidation. a** Schematic of the decoupled system with p-VHCF mediator electrode for H₂ production and hydrazine oxidation with electricity cogeneration. **b** CV curve of the p-VHCF electrode (black line), the LSV curves of RuO₂/IrO₂ electrode in 6 M H₂SO₄ electrolyte (red line) and Pt mesh electrode in 0.5 M H₂SO₄ + 0.1 M N₂H₄ electrolyte (purple line). **c** Chronopotentiometry curve of the decoupled system at a current of 100 mA for H₂ generation and at a current of 10 mA for hydrazine oxidation with a charge/discharge capacity of 2.78 mAh. **d** Practical amounts of obtained H₂ and N₂, where the Step 1 was performed at 20 mA for 2000 s and Step 2′ at 10 mA for 4000 s. **e** Gas chromatography data for HzOR (the insert is the V-t curve of the Step 2′ during GC test at 20 mA). **f** The Polarization and power density curves of p-VHCF-N₂H₄ battery. **g** Schematic of decoupled electrolysis system driven by solar cell in Step 1 for high-rate H₂ production at day-time, followed by the hydrazine oxidation with electricity generation in Step 2′ at night-time. **h** The I-t curve of the Step 1 driven by Si solar cell. **i** The V-t curve of the Step 2′ with a discharge current of 1 mA.

Figure 6b shows the comparison of the LSV curves of OER in 6 M H₂SO₄ electrolyte and HzOR in 0.5 M H₂SO₄ + 0.1 M N₂H₄ electrolyte, where the onset potential of HzOR locates at 0 V (vs. Ag/AgCl), more negative than the onset potential of OER and apparently lower than the redox potential of p-VHCF. Thus, the reduction of p-VHCF$_{ox}$ and hydrazine oxidation is spontaneous if only considering the thermodynamic potential of the electrodes. It seems that a p-VHCF-N₂H₄ liquid battery can be constructed in Step 2′. To reveal this point, the chronopotentiometry data of Step 1 at 100 mA and Step 2′ at 10 mA with the same charge capacity of 2.78 mAh were tested (Fig. 6c). Clearly, Step 1 exhibits a cell voltage of about 1.1 V for hydrogen production, where the p-VHCF mediator is charged to 1.0 V (vs. Ag/AgCl). Due to the potential difference between the cathodic reduction of p-VHCF and the anodic HzOR, the Step 2′ displays a discharge voltage of about 0.5 V with spontaneous nitrogen gas production (Supplementary Movie 2), indicating the successful formation of a p-VHCF-N₂H₄ liquid battery. The rate capability of the p-VHCF-N₂H₄ battery at different discharge currents was also tested (Supplementary Fig. 20).

Even at the high discharge current of 20 mA, the battery still can exhibit an average voltage of about 0.35 V with 100% Coulombic efficiency. The formed p-VHCF-N₂H₄ battery can deliver an open-circuit voltage about 0.84 V. When the p-VHCF electrode is charged to 1.2 V (vs. Ag/AgCl), the battery can deliver an open-circuit voltage of 1.05 V (Supplementary Fig. 21). The aqueous batteries, especially aqueous proton batteries with only the proton charge carriers, usually hold an electrochemical window of 0-1.2 V, but those secondary batteries intrinsically own the perceived merits of high safety, low cost, easy manufacture, fast kinetics, and long-term cycling stability[43,44,52–56]. The gas products and purity were also measured and the data are shown in Fig. 6d-e, indicating a nearly 100% Faraday efficiency for removing hydrazine to produce innocuous N₂ from acid waste water. Furthermore, we summarized the polarization and power density curves of the p-VHCF-N₂H₄ battery (Fig. 6f). A maximum power of around 9 mW is obtained, which is close to some reported liquid fuel batteries[24,51].

More importantly, this decoupled system is proposed to enable the flexible energy conversion and storage. We can use solar energy to

drive the Step 1 process for high-rate $H_2$ production at day-time, and achieve hydrazine oxidation with electricity generation (Step 2′) through the p-VHCF-$N_2H_4$ liquid battery at night-time (Fig. 6g). To clarify this point, we built a Si solar cell driven decoupled electrolysis system and tested the performances of these two separate processes. As shown in Fig. 6h, the operating current of the solar cell driven Step 1 is around 22 mA, which matches with the value estimated from the intersection of the LSV curve of the Step 1 for $H_2$ production and the current-voltage curve of the Si solar cell (Supplementary Fig. 22). Meanwhile, the Step 2′ can output stable electricity (Fig. 6i). In this regard, we believe this design shows predictable potential for flexible energy conversion and storage compared with the direct hydrazine electrolysis (Supplementary Fig. 23).

In summary, we have successfully prepared pre-protonated vanadium hexacyanoferrate (p-VHCF) Prussian blue analog as a solid-state redox mediator for the decoupled electrolysis systems. Due to the enhanced hydrogen bonding network, this electrode delivers a high reversible specific capacity up to 128 mAh g$^{-1}$ and long cycling performance of 6000 cycles with capacity retention about 100% at a current density of 10 A g$^{-1}$. The p-VHCF electrode also shows good performances in various acid electrolytes, demonstrating the promising potential in decoupled water electrolysis, batteries, and other electrochemical applications. Most importantly, a decoupled electrolysis system for hydrogen production and hydrazine oxidization is built, which realizes separate $H_2$ generation, electrical energy storage, and green treatment of hydrazine hazards. Typically, solar energy can be used to drive the $H_2$ production at day-time. The hydrazine oxidation with electricity generation can be achieved through the p-VHCF-$N_2H_4$ liquid battery at night-time. This architecture shows a promising solution to facilitate renewables-to-hydrogen conversion, and provides new ideas to build the hybrid energy conversion/storage system.

## Methods

### Sample fabrication

Vanadium precursor solution was synthesized via a chemical reduction method according to the pervious report[33]. Typically, 50 ml of 32% HCl was diluted with deionized water to 75 ml, and 4 g $V_2O_5$ powder was then added to the HCl solution to form a yellow suspension. Afterwards, 700 µl glycerol was added dropwise, and the solution was continuously stirred for 1 h to form a clear blue solution at 60 °C.

The vanadium hexacyanoferrate (VHCF) was prepared by a co-precipitation method. Specifically, the as-prepared vanadium precursor solution (9.375 ml) was diluted with deionized water to 50 ml under stirring to form a transparent blue solution. Then, 50 mL $K_3Fe(CN)_6$ solution (0.072 M) was dropwise added into above solution and stirred for 9 hours under 60 °C. Finally, the green powder was collected after centrifugation, washing, and vacuum drying overnight.

The protonated vanadium hexacyanoferrate (p-VHCF) was obtained by chemical reduction method[32,34]. Typically, 0.2 g VHCF powder was ultrasonically dispersed into 20 mL of deionized water. Then, 10 mL of hydrazine hydrate (0.05 M) was added to the above suspension and stirred for 2 h under $N_2$ atmosphere. Finally, the suspension was centrifuged and the precipitate was washed with deionized water repeatedly. The obtained yellow powder was dried in a 60 °C vacuum oven overnight to obtain p-VHCF powder.

### Characterization

X-ray diffraction patterns (XRD) were obtained from a PANalytical X'pert MPD Pro diffractometer using Ni-filtered Cu Kα irradiation. The morphology and elemental analysis were studied using a JEOL JSM-7800F field emission scanning electron microscope (SEM) combined with an energy dispersive spectroscopy (EDS) detector. Transmission electron microscopy (TEM) images were obtained using JEOL JEM-2100Plus. Fourier transform infrared (FTIR) spectroscopy was recorded using KBr pellets on a Bruker Vertex 70 FTIR spectrometer in the

wavenumber range of 400-4000 cm$^{-1}$. The thermogravimetric analysis (TGA 209 F1) tests were carried out under Ar atmosphere at a ramp rate of 10 °C min$^{-1}$. X-ray Photoelectron Spectroscopy (Thermo Fisher ESCALAB) analysis was used to determine their chemical compositions. The in-situ Raman measurement was conducted in the wavenumber range of 700-2300 cm$^{-1}$ using a DXR Raman microscope (excitation length: 532 nm) with real-time CV at 0 to 1.2 V (vs. Ag/AgCl) and a 10 s exposure length.

### Electrode preparation and electrochemical tests

The p-VHCF electrode was prepared by mixing p-VHCF powder, acetylene black, and polytetrafluoroethylene binder in a mass ratio of 70:20:10 in a mortar. Then, a few drops of isopropanol were added into the mixture under stirring until the mixture formed a homogeneous slurry. The slurry mixtures were then rolled into a film and pressed onto a Ti-mesh based current collector to form the p-VHCF electrode with a mass loading of about 2 - 2.5 mg cm$^{-2}$. Cyclic voltammetry (CV) and linear sweep voltammetry (LSV) measurements were measured in 6 M $H_2SO_4$ electrolyte with a typical three-electrode system, where the counter and reference electrodes were Pt plate electrode and Ag/AgCl (saturated KCl) electrode, respectively. All above tests were performed on an electrochemical workstation (CHI 760E, Shanghai Chenhua, China). The galvanostatic charge-discharge was evaluated in the voltage range of 0-1.2 V (vs. Ag/AgCl) using a LAND test system using the above mentioned three-electrode mode.

The decoupled acid water electrolyzer includes the HER electrode (commercial Pt-coated Ti-mesh electrode, 1.5 × 2 cm$^2$), the OER electrode (commercial $RuO_2$/$IrO_2$ coated Ti-mesh electrode, 1.5 × 2 cm$^2$), and the p-VHCF mediator electrode (mass loading is 60 mg cm$^{-2}$, 1.5 × 2 cm$^2$) in the acid electrolyte (6 M $H_2SO_4$). The image of the decoupled water electrolysis cell is shown in Supplementary Fig. 11, where the p-VHCF electrode is located between the HER electrode and OER electrode without using any membrane in the cell. The decoupled electrolyzer for hydrogen production and hydrazine oxidation includes the HER electrode (commercial Pt-coated Ti-mesh electrode, 1.5 × 2 cm$^2$), the HzOR electrode (commercial Pt-coated Ti-mesh electrode, 1.5 × 2 cm$^2$), and the p-VHCF mediator electrode (mass loading is 60 mg cm$^{-2}$, 1.5 × 2 cm$^2$). The HER electrode and p-VHCF electrode are located in 6 M $H_2SO_4$ electrolyte and the HzOR electrode is located in 0.5 M $H_2SO_4$ + 0.1 M $N_2H_4$ electrolyte. The two electrolytes are separated by a proton exchange membrane.

The decoupled electrolysis performance was investigated using the chronopotentiometry method. In Step 1, the p-VHCF electrode and HER electrode act as the anode and cathode, respectively. In Step 2 (or Step 2′), the OER (or HzOR) electrode and oxidized p-VHCF electrode act as the anode and cathode, respectively. The cell voltages of Step 1 and Step 2 (Step 2′) were used to characterize the decoupled electrolysis properties. The chronopotentiometry data of the single electrode (the p-VHCF electrode, the OER/HzOR electrode, or the HER electrode) were also detected using an Ag/AgCl electrode as the reference electrode during Step 1 and Step 2 (Step 2′). A CHI 760E electrochemical workstation was used for the electrochemical measurement. The gas purity was detected by GC measurement with Ar gas as carrier gas.

## Data availability

The authors declare that the data supporting the findings of this study are available within the paper and its Supplementary Information files. Source data are provided with this paper. Additionally, the corresponding authors can provide the raw data upon request. Source data are provided with this paper.

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

## Acknowledgements

L.G. and Y.C. acknowledge the financial support from the National Natural Science Foundation of China (No. 51888103). Y.C. acknowledges the financial support from the National Natural Science Foundation of China (No. 52076177), China National Key Research and Development Plan Project (No. 2021YFF0500503), Key Research and Development Program of Shaanxi (No. 2024GH-YBXM-02), and China Fundamental Research Funds for the Central Universities (No. xpt022022005).

## Author contributions

F.L. and Y.C. conceived and designed the research. F.L. synthesized and characterized the samples, and performed electrochemical experiments. J.W., X.L., Z.Z., L.P., X.Z., and L.G. assisted in sample characterizations and electrochemical tests. F.L., J.W., L.G., and Y.C. analyzed the experimental results and wrote the manuscript. Y.C. supervised the entire research work. All authors read the manuscript and contributed to the discussion of the results.

## Competing interests

The authors declare no competing interests.

## Additional information

**Peer review information** : *Nature Communications* thanks Jianping Yang, Kazuhiro Takanabe and the other, anonymous, reviewer(s) for their contribution to the peer review of this work. A peer review file is available.

