## [Peer review file · Nature Communications]

REVIEWER COMMENTS

Reviewer #1 (Remarks to the Author):

This study reports redox material that can be applied for decoupled water electrolysis. The developed material is solid state vanadium hexacyanoferrate. The material shows sufficient tolerance in high current density operation up to 10A/g, and more interestingly high cyclability. Compared to other materials reported in a similar condition, it is outstanding that the performance remains unchanged after 6000 cycles. The material is therefore interesting to be reported, but the scientific narrative and characterization, especially compared to similar articles, are rather limited. Therefore, this reviewer recommends more characterization before considering the publication in esteemed Nature Communications.

First, the assignment of redox cycles is insufficient. What about crystal structure during redox actions? From ex-situ XPS, it is unclear whether three redox events (or even more) are ascribed to vanadium or iron, or both (but then which events correspond to which peaks). Is there any access to in situ XRD or EXAFS?

Second, it is good that the reaction is operated with stable region of the electrochemistry. It would be useful to know the limit of this material by applying faster capacity measurement, and high current density. CV outside the redox window to show the material decomposition would also help to decide the operation region.

Third, hydrogen evolution as a function of current density is interesting to be reported.

Forth, the potential of redox mediator as a function of time at different current densities is also interesting to investigate.

Overall, the high cyclability under concentrated phosphoric acid as a solid redox mediator is interesting. Further characterization of the sample makes the paper more attractive and make it worth publication in Nature Communications.

Reviewer #2 (Remarks to the Author):

The authors have conducted a compelling investigation into the use of pre-protonated vanadium hexacyanoferrate (H-VHCF) as a redox mediator to separate the hydrogen evolution reaction (HER) and oxygen evolution reaction (OER) in water splitting. In the manuscript, they demonstrate the generation of hydrogen gas and the versatile applications of charged H-VHCF, which can be employed either for oxygen gas production through electrolyzer anode utilization or for electricity generation when coupled with hydrazine oxidation. This combination of a redox mediator (H-VHCF) with hydrazine oxidation is an intriguing concept. Furthermore, the H-VHCF electrode exhibits remarkable reversible specific capacity and outstanding cycling stability. As a result, this manuscript is likely to garner significant attention in the field of energy conversion and storage research, making it a suitable candidate for publication in Nature Communications. Nevertheless, there are several points that should be addressed before recommending it for publication:

1. In Figure 3g, the XRD patterns of the H-VHCF material at "c" and "g" states display distinct peak information in the 50°-60° region. The authors should provide a reasonable explanation for this discrepancy.

2. In Figure 4, the charge/discharge times of the H-VHCF electrode are set at 10 minutes, which may appear too short to effectively buffer fluctuations from renewable sources. The authors should investigate the performance of the redox materials over long-term charge/discharge processes. Additionally, it would be valuable to assess the device's efficiency when the H-VHCF material is partially charged and discharged.

3. Figure 5c presents charge/discharge characteristics of the H-VHCF electrode that differ from the results in Figures 4a and 4b. The authors should provide commentary for readers to help understand these differences.

4. The authors state that the H-VHCF electrode exhibits excellent cycling performance, with 6000 cycles and 100% capacity retention at a high current density of 10 A/g. However, it would be beneficial to include data on capacity retention at a lower charge/discharge rate to provide a more comprehensive view of the material's performance.
5. The abbreviation "H-VHCF" for pre-protonated vanadium hexacyanoferrate may be confusing for readers and does not clearly convey information about the material. The authors should select a more concise and transparent abbreviation.
6. The manuscript's title is somewhat inaccurate and could be misleading. The electrochemical tests in the manuscript encompass two distinct parts: decoupling water splitting using H-VHCF as a redox mediator and coupling two half-reactions for electricity generation. Therefore, the statement "Decoupled Water/Hydrazine Electrolysis" is inappropriate and should be revised to accurately reflect the content.
7. Additionally, the manuscript should include a schematic representation of the coupling strategy for the reduction process of H-VHCF and hydrazine oxidation in the Scheme 1 to form a new electrolysis architecture, as this is a crucial aspect of the research that should be visually represented.

Reviewer #3 (Remarks to the Author):

This work presents a decoupled acid water electrolysis system by a preprotonated vanadium hexacyanoferrate (H-VHCF) solid-state redox mediator. Although the work is well done and the electrochemistry sound, I feel the work lacks novelty and does not constitute new science that is of interest to a broad readership targeted by Nature Communications. I think this should be published in a more specialist journal.

1. This decoupled architecture was performed in 6 M H₂SO₄ electrolyte. For conventional acid water electrolysis, the electrolyte concentration is much lower than 6 M H₂SO₄, such as 0.5 M H₂SO₄, 1 M H₂SO₄, 0.1 M HClO₄. The high concentration of the electrolyte will put greater demands for electrolyzers and electrodes.
2. There is no detailed introduction about why the authors using the hydrazine oxidation reaction (HzOR) to replace OER, the application of HzOR.
3. The redox mediator and the decoupled process of this work is similar to the reported work of Adv. Energy Mater. 2021, 11, 2102057. Based on the Grotthuss proton conduction, the rate performance of H-VHCF is inferior to that of CuFe-TBA.
4. Please provide the structure and morphology change, such as the XRD, XPS, and SEM images, after long cycling.
5. From Figure 5b, the oxidation potential of HzOR is close to the reduction potential of HER, in practical it will consume low energy to produce hydrogen directly by hydrazine electrolysis.
6. For the decoupled hydrazine electrolysis process, the authors indicate that this process could remove hydrazine from acid waste water. Owing to the limited capacity of H-VHCF, how long will it take to remove hydrazine to a safe concentration? In addition, it is meaningless to employ this process as a secondary battery due to its lower discharge voltage.
7. In page 11 and 12, the unit of the electrode area should be cm².
8. The format of references should be unified.

Response to the Reviewer #1

General comment: This study reports redox material that can be applied for decoupled water electrolysis. The developed material is solid state vanadium hexacyanoferrate. The material shows sufficient tolerance in high current density operation up to 10A/g, and more interestingly high cyclability. Compared to other materials reported in a similar condition, it is outstanding that the performance remains unchanged after 6000 cycles. The material is therefore interesting to be reported, but the scientific narrative and characterization, especially compared to similar articles, are rather limited. Therefore, this reviewer recommends more characterization before considering the publication in esteemed Nature Communications.

Response: Thank you very much for kindly reviewing our manuscript. Your thorough review is really helpful to improve our manuscript. We have added more characterization results and discussion according to the reviewers' insightful comments, and made the revisions according to your suggestions.

Comment 1: The assignment of redox cycles is insufficient. What about crystal structure during redox actions? From ex-situ XPS, it is unclear whether three redox events (or even more) are ascribed to vanadium or iron, or both (but then which events correspond to which peaks). Is there any access to in situ XRD or EXAFS?

Response: Thanks for your insightful comment. It's hard for us to find the suitable devices to perform the in-situ XRD or EXAFS measurement because the required light-window (beryllium window) of the in-situ devices is not acid resistant. However, we have performed in-situ Raman tests and more detailed ex-situ measurements (ex-situ XRD and XPS) to reveal the structure and composition changes during the redox reactions.

Firstly, the ex-situ XRD data of the redox mediator charged/discharged under varied potentials (including three pairs of redox peaks) were collected and shown in **Fig. 3f-g**. Clearly, there is no new peak appearing or original peak disappearing, indicating a non-phase transition process during the redox actions. Only the reversible lattice shrinkage and expansion occurred due to ion radius change during the proton insertion and de-insertion. Because the redox mediator is stable in the air, we think the detailed ex-situ XRD results at different potentials can reveal the crystal structures during redox actions according to the literatures [*Refs. Adv. Energy Mater.* 2021, 2102057; *Joule* 2021, 5, 149–165; *ChemSusChem.* 2023, 16, e202201689].

Secondly, the in-situ Raman and ex-situ XPS measurements were performed to illustrate the redox centers. As shown in the in-situ Raman spectra (**Fig. 4a-b**), the peak of V=O group located at 892 cm⁻¹ becomes gradually weakened during the charging process, while the peak intensity slowly increases

during the discharging process. In addition, the peak position and intensity of C≡N groups bonded to iron ions located at 2110 and 2157 cm⁻¹ also changes reversibly. Typically, the peaks slightly shift toward higher wavenumber and then reverses backward during the charging and discharging cycle, indicating the redox of Fe²⁺ and Fe³⁺ ions. [Refs. *Chem. Mater.* 2015, 27, 1997; *Adv. Funct. Mater.* 2017, 27, 1604307; *Angew. Chem. Int. Ed.* 2022, 61, e202205867].

Ex-situ XPS measurement was further carried out to track the chemical status of components at various charging/discharging states (**Fig. 4c-h**). The partial V undergoes a shift of V³⁺ → V⁴⁺ → V⁵⁺ during charging process from initial 0.52 V to 1.2 V (vs. Ag/AgCl) and reserves back during the discharging process. This further indicates that V ions take part in the electrochemical redox reaction (**Fig. 4c**). According to the valence distribution diagram of V ions, the corresponding ratios of V⁴⁺/(V³⁺+ V⁴⁺) and V⁵⁺/(V⁴⁺+ V⁵⁺) are apparently increased at 0.6 V and 1.2 V (vs. Ag/AgCl), implying that the oxidation of V³⁺ to V⁴⁺ starts at the O₁ peak in the CV curve, and the oxidation of V⁴⁺ to V⁵⁺ starts at the O₃ peak in the CV curve (**Fig. 4f and h**). Meanwhile, the valance distribution ratio of Fe³⁺/(Fe²⁺+Fe³⁺) is significantly increased at 0.9 V during the charging process, indicating the oxidation of Fe²⁺ to Fe³⁺ starting at the O₂ peak in the CV curve (**Fig. 4d and g**). Therefore, the three redox peaks of O₁/R₁, O₂/R₂, O₃/R₃ in the CV curve of the p-VHCF electrode (**Fig. 4e**) should respectively corresponded to the redox reactions of V³⁺/V⁴⁺, Fe²⁺/Fe³⁺, and V⁴⁺/V⁵⁺, consistent with those results reported in the literature. [Refs. *Chem. Commun.*, 2020, 56, 11803-11806; *J. Power Sources* 2021, 484, 229263].

[Revision]: [Manuscript Page 11-14]

The ex-situ XRD patterns of the p-VHCF electrode at different charge/discharge states are presented in Fig. 3f-g. Clearly, there is no new peak appearing or original peak disappearing, indicating a non-phase transition process. While during the charging process with the extraction of protons (from point 1 to 5 in Fig. 3f), the (200) and (400) diffraction peaks shift to lower degrees, implying an increase of the lattice parameter. Likewise, upon the discharge process (from point 5 to 9 in Fig. 3f), these diffraction peaks shift back to original states, revealing high reversibility of the electrode structure.

In addition, the in-situ Raman and ex-situ XPS measurements were performed to illustrate the redox centers. As shown in the in-situ Raman spectra (Fig. 4a-b), the peak of V=O group located at 892 cm⁻¹ becomes gradually weakened during the charging process, while the peak intensity slowly increases during the discharging process. The peak position and intensity of C≡N groups bonded to iron ions located at 2110 and 2157 cm⁻¹ also change reversibly. Typically, the peaks slightly shift toward higher wavenumber and then reverses backward during the charging and discharging cycle,

indicating the redox of Fe^{2+} and Fe^{3+} ions⁴⁷⁻⁴⁹. These results confirm that the $\text{V}=\text{O}$ and $[\text{Fe}(\text{CN})_6]^{4-}$ are the active redox sites during the redox process.

Ex-situ XPS measurement was further carried out to track the chemical status of components at various charging/discharging states (Fig. 4c-h). The partial V undergoes a shift of $\text{V}^{3+} \rightarrow \text{V}^{4+} \rightarrow \text{V}^{5+}$ during charging process from initial 0.52 V to 1.2 V (vs. Ag/AgCl) and reserves back during the discharging process. This further indicates that V ions take part in the electrochemical redox reaction (Fig. 4c). According to the valence distribution diagram of V ions, the corresponding ratios of $\text{V}^{4+}/(\text{V}^{3+} + \text{V}^{4+})$ and $\text{V}^{5+}/(\text{V}^{4+} + \text{V}^{5+})$ are apparently increased at 0.6 V and 1.2 V, implying that the oxidation of V^{3+} to V^{4+} starts at the O_1 peak in the CV curve, and the oxidation of V^{4+} to V^{5+} starts at the O_3 peak in the CV curve (Fig. 4f and h). Meanwhile, the valence distribution ratio of $\text{Fe}^{3+}/(\text{Fe}^{2+} + \text{Fe}^{3+})$ is significantly increased at 0.9 V during the charging process, indicating the oxidation of Fe^{2+} to Fe^{3+} starting at the O_2 peak in the CV curve (Fig. 4d and g). Therefore, the three redox peaks of O_1/R_1 , O_2/R_2 , O_3/R_3 in the CV curve of the p-VHCF electrode (Fig. 4e) should respectively corresponded to the redox reactions of $\text{V}^{3+}/\text{V}^{4+}$, $\text{Fe}^{2+}/\text{Fe}^{3+}$, and $\text{V}^{4+}/\text{V}^{5+}$, consistent with those results reported in the literature^{33,46}.

Fig. 3 | **f** Charge/discharge potential profiles of the p-VHCF electrode (Points 2-8 correspond to the potentials of the three pairs of redox peaks in the CV curve). **g** Corresponding ex-situ XRD patterns at different charging/discharging states and the enlarged views of the (200) and (400) peaks in the XRD patterns.

Fig. 4 | Ex-situ/in-situ structure and composition measurement. **a-b** In-situ Raman line spectra and 2D contour plots of the p-VHCF electrode. Ex-situ XPS spectra at different charging and discharging states. **c** V 2p and **d** Fe 2p spectra of the p-VHCF electrode. **e** CV curve of the p-VHCF electrode. **f-g** Valence distribution ratio diagrams of V and Fe of the p-VHCF electrode during the redox reaction.

Comment 2: It is good that the reaction is operated with stable region of the electrochemistry. It would be useful to know the limit of this material by applying faster capacity measurement, and high current density. CV outside the redox window to show the material decomposition would also help to decide the operation region.

Response: Thanks for the comment. We have tested the limit of this material by applying faster capacity measurement and higher current density (**Fig. 3c**). It is found that the p-VHCF electrode can achieve an acceptable capacity of 47 mAh g⁻¹ at a current density of 200 A g⁻¹. When the current density is up to 300 A g⁻¹, the electrode can only deliver a capacity of 17 mAh g⁻¹, indicating the limit of this electrode.

In addition, CV electrochemical window of the p-VHCF electrode was further changed from the potential range of 0-1.2 V to -0.23-1.47 V (vs. Ag/AgCl), which is shown in **Fig. R1**. Clearly, there are no extra redox peaks in this wider range. However, the hydrogen evolution reaction and oxygen evolution reaction occur on this electrode at -0.18 V and 1.36 V (vs. Ag/AgCl), respectively, indicating the side reactions of this material. Therefore, we chose the potential region of 0-1.2 V (vs. Ag/AgCl) as the operation region in the current work.

Fig. R1 CV curve of the p-VHCF electrode in the potential range of -0.23-1.47 V (vs. Ag/AgCl).

[Revision]: [Manuscript Page 9, Fig. 3c]

Even at a high current density of 200 A g⁻¹, a respectable capacity of 47 mAh g⁻¹ was still achieved, indicating ultrafast proton insertion/de-insertion rate. When the current density was up to 300 A g⁻¹, the electrode only delivers a capacity of 17 mAh g⁻¹, indicating the limit of this electrode.

[Manuscript Page 11, Fig. 3c]

Fig. 3 | c Rate performance of the p-VHCF electrode for selected current densities.

Comment 3: Hydrogen evolution as a function of current density is interesting to be reported.

Response: Thanks for the comment. Hydrogen evolution as a function of varied current was obtained. As shown in **Supplementary Fig. 18**, with all step time of 800 s, the hydrogen production amount was increased with the increased current. Typically, this decoupled system achieves Faraday efficiencies of higher than 99% for hydrogen evolution at various current densities.

[Revision]: [Manuscript Page 16]

Meanwhile, hydrogen evolution as a function of varied current was obtained. As shown in **Supplementary Fig. 18**, with all step time of 800 s, the hydrogen production amount was increased with the increased current. Typically, this decoupled system achieves Faraday efficiencies of higher than 99% for hydrogen evolution at various currents.

[SI Page S10, Supplementary Fig. 18]

Supplementary Fig. 18 Hydrogen evolution as a function of current with all step time of 800 s.

Comment 4: The potential of redox mediator as a function of time at different current densities is also interesting to investigate.

Response: Thanks for the suggestion. The potential of redox mediator as a function of time at different currents is given in **Supplementary Fig. 14**. The polarization gradually increases with the increase of applied current from 5 to 100 mA, with the coulombic efficiency of 98% even at a high current of 100 mA.

[Revision]: [Manuscript Page 15-16]

The potential of the p-VHCF electrode as a function of time was acquired at different currents, as shown in **Supplementary Fig. 14**. The polarization gradually increases with the increase of applied

current from 5 to 100 mA, with the coulombic efficiency of 98% even at a high current of 100 mA.

[Revision]: [SI Page S8, Supplementary Fig. 14]

Supplementary Fig. 14 The potential of redox mediator as a function of time at different currents (the mass loading of the p-VHCF is 60 mg cm^{-2}).

Comment 5: Overall, the high cyclability under concentrated phosphoric acid as a solid redox mediator is interesting. Further characterization of the sample makes the paper more attractive and make it worth publication in Nature Communications.

Response: Thank you very much for your helpful suggestions. We have added more characterizations to improve the manuscript. Typically, we added one more figure in the main text (Fig. 4) and several images in the supplementary information. In addition, we tested the cyclability under concentrated phosphoric acid (10 M) according to the literature [*Ref. Adv. Funct. Mater.*2023, 33, 2214466]. As shown in **Supplementary Fig. 10a**, this material also shows excellent cycling stability and a reservable redox capacity of 98 mAh g^{-1} in 10 M H_3PO_4 . It is reported that the freezing point of H_3PO_4 aqueous solution can be lower than $-80 \text{ }^\circ\text{C}$ with the increase of electrolyte concentration to 9 M. This result indicates that the p-VHCF mediator has a potential application at ultralow temperatures for H_2 production [*Refs. J. Am. Chem. Soc.* 2021, 143, 20302–20308; *Adv. Energy Mater.* 2020, 10, 2000968]. Moreover, this electrode is also stable in 2 M H_3PO_4 electrolyte **Supplementary Fig. 10b**.

[Revision]:

[Manuscript Page 10-11]:

In addition, we tested the performances of the p-VHCF electrode in 10 M and 2 M H_3PO_4 electrolytes. As shown in Supplementary Fig. 10a, this material shows excellent cycling stability and a redox capacity of 98.8 mAh g^{-1} in 10 M H_3PO_4 . It is reported that the freezing point of H_3PO_4 electrolyte can be lower than $-80 \text{ }^\circ\text{C}$ with the increase of electrolyte concentration to 9 M. This result

indicates that the p-VHCF mediator has a potential application at ultralow temperatures for H₂ production^{34,44}. Moreover, this electrode is also stable in 2 M H₃PO₄ electrolyte (Supplementary Fig. 10b). Overall, owing to the good performances in different electrolytes, the p-VHCF electrode seems promising in decoupled electrolysis, batteries, and other electrochemical applications.

[SI Page S6, Supplementary Fig. 10]:

Supplementary Fig. 10 Cycling performance of the p-VHCF electrode in (a) 10 M and (b) 2 M H₃PO₄ electrolyte at 10 A g⁻¹.

Response to the Reviewer #2

General Comment: The authors have conducted a compelling investigation into the use of pre-protonated vanadium hexacyanoferrate (H-VHCF) as a redox mediator to separate the hydrogen evolution reaction (HER) and oxygen evolution reaction (OER) in water splitting. In the manuscript, they demonstrate the generation of hydrogen gas and the versatile applications of charged H-VHCF, which can be employed either for oxygen gas production through electrolyzer anode utilization or for electricity generation when coupled with hydrazine oxidation. This combination of a redox mediator (H-VHCF) with hydrazine oxidation is an intriguing concept. Furthermore, the H-VHCF electrode exhibits remarkable reversible specific capacity and outstanding cycling stability. As a result, this manuscript is likely to garner significant attention in the field of energy conversion and storage research, making it a suitable candidate for publication in Nature Communications. Nevertheless, there are several points that should be addressed before recommending it for publication:

Response: We appreciate the positive evaluation of our work and the valuable suggestions. Your thorough review has been instrumental in improving our manuscript. We have carefully considered your concerns and made the revisions according to your suggestions.

Comment 1: In Figure 3g, the XRD patterns of the H-VHCF material at "c" and "g" states display distinct peak information in the 50°-60° region. The authors should provide a reasonable explanation for this discrepancy.

Response: Thank you for this comment. The previous XRD data had a poor signal-to-noise ratio, so the XRD spectra have been retested more precisely. As shown in the **Fig. 3f-g**, there is no distinct peak information in the 50°-60° region and no new peak appearing or original peak disappearing, indicating a non-phase transition process. During the charging process with the extraction of protons, the (200) and (400) diffraction peaks shift to lower degrees, implying an increase of the lattice parameter. Likewise, upon the discharge process, these diffraction peaks shift back to original states, revealing high reversibility of the electrode structure.

[Revision]:

[Manuscript Page 12, Fig. 3f-g]:

Fig. 3 | **f** Charge/discharge potential profiles of the p-VHCF electrode (Points 2-8 correspond to the potentials of the three pairs of redox peaks in the CV curve). **g** Corresponding ex-situ XRD patterns at different charging/discharging states and the enlarged views of the (200) and (400) peaks in the XRD patterns.

Comment 2: In Figure 4, the charge/discharge times of the H-VHCF electrode are set at 10 minutes, which may appear too short to effectively buffer fluctuations from renewable sources. The authors should investigate the performance of the redox materials over long-term charge/discharge processes. Additionally, it would be valuable to assess the device's efficiency when the H-VHCF material is partially charged and discharged.

Response: We appreciate this insightful comment. In the decoupled system, we used a short time for the facile comparison of different reaction parameters. As the reviewer mentioned, it is meaningful to examine the long-term performance for buffering fluctuations from renewable sources. The operation time and current can be adjusted by changing the mass loading of mediator electrode. [Refs. *Angew. Chem. Int. Ed.* 2018, 57, 2904–2908; *Angew. Chem. Int. Ed.* 2019, 58, 4622–4626; *Angew. Chem. Int. Ed.* 2023, e202303563]. Therefore, we increased the mass loading of the p-VHCF electrode to 466 mg cm⁻², and the step time of 12 h for hydrogen or oxygen production could be achieved, indicating the good performance of the redox materials over long-term charge/discharge processes (**Supplementary Fig. 15**).

In addition, the efficiency of the decoupled water electrolysis device has been given in the revised manuscript. According to the previous reports [Refs. *Nat. Chem* 2013, 5, 403-409; *Nat. Commun.* 2016, 7, 11741], the efficiency of the two-step system can be calculated by comparing its total driven voltage (Step 1 + Step 2) to the driven voltage of the corresponding one-step system. In our system, the

chronopotentiometry curve tested at 100 mA of the two-step system is shown in **Supplementary Fig. 13a**, where it can be observed that the two steps display a total cell voltage of 1.84 V (1.0 + 0.84 V). The achieved chronopotentiometry curve of one-step electrolysis with the same gas evolution electrodes is shown in **Supplementary Fig. 13b**, where it can be detected that the cell exhibits a voltage of 1.8 V with the applied current of 100 mA. Therefore, the efficiency of the decoupled cell was 97.8% (1.8/1.84) compared to the corresponding one-step system.

[Revision]:

[Manuscript Page 16]:

Noticed that the electrolytic step time can be easily managed by varying the loading amount of redox mediator. When the mass loading of the p-VHCF was increased from 60 mg cm⁻² to 466 mg cm⁻², the electrolytic cell can be cycled with a step-time of 12 h, indicating the good performance of the mediator material over long-term charge/discharge processes (Supplementary Fig. 15).

[SI Page S9, Supplementary Fig. 15]:

Supplementary Fig. 15 Chronopotentiometry curve of the decoupled electrolyzer with a step-time of 12 h.

[Manuscript Page 15]:

The efficiency of the decoupled system is calculated to be 97.8% at 100 mA compared to the corresponding one-step system according to the previous reports (Supplementary Fig. 13)^{13,50}.

[SI Page S8, Supplementary Fig. 13]:

Supplementary Fig. 13 Voltage comparison at 100 mA between (a) two-step decoupled electrolysis and (b) one-step direct electrolysis without the membrane.

As shown in Supplementary Fig. 13a, the two steps display a total cell voltage of 1.84 V (1.0 + 0.84 V). The chronopotentiometry curve of one-step electrolysis with the same HER/OER electrodes is shown in Supplementary Fig. 13b, where the cell exhibits a voltage of 1.8 V with the applied current of 100 mA. Therefore, the efficiency of the decoupled cell was 97.8% (1.8/1.84) compared to the corresponding one-step system.

Comment 3: Figure 5c presents charge/discharge characteristics of the H-VHCF electrode that differ from the results in Figures 4a and 4b. The authors should provide commentary for readers to help understand these differences.

Response: In our revised manuscript, because we added one more figure, the previous Fig. 5c, Fig. 4a and 4b are Fig. 6c, Fig. 5a and 5b.

Fig. 5a and 5b (previous Fig. 4a and 4b) show the decoupled water electrolysis performances, where the two processes (Step 1 and Step 2) were performed at the same currents (5 mA in Fig. 5a and 100 mA in Fig. 5b) with a certain step time. That means the p-VHCF electrode is **charged/discharged at a same rate**. The Fig. 6c (previous Fig. 5c) involves an electrolysis process of Step 1 for H₂ production and a battery discharge process of Step 2' for hydrazine oxidation. These two processes were performed at different currents with the same charges (e.g., Step 1 at 100 mA, Step 2' at 10 mA, charge and discharge capacities are both 2.78 mA h), which means the p-VHCF electrode is **charged at a high rate and discharged at a low rate**. Therefore, the charge/discharge characteristics of the p-VHCF electrode illustrated in Fig. 6c are kind of different. Specifically, the charge characteristic of the p-VHCF electrode in the Step 1 of Fig. 5b is similar to that in the Step 1 of Fig. 6c, because they have the same currents of 100 mA for hydrogen production. As suggested, we compared those charge

characteristics of the p-VHCF electrode shown in Fig. R2. Also, we provided a commentary in the Revised Manuscript.

Fig. R2 Chronopotentiometry curves in previous (a) Fig. 4b and (b) Fig. 5c.

[Revision]:

[Manuscript Page 19]:

To reveal this point, the chronopotentiometry data of Step 1 at 100 mA and Step 2' at 10 mA with the same charge capacity of 2.78 mA h were tested (Fig. 6c).

Comment 4: The authors state that the H-VHCF electrode exhibits excellent cycling performance, with 6000 cycles and 100% capacity retention at a high current density of 10 A/g. However, it would be beneficial to include data on capacity retention at a lower charge/discharge rate to provide a more comprehensive view of the material's performance.

Response: Thanks for your valuable suggestion. We have tested the capacity retention at a lower charge/discharge rate of 2 A g⁻¹. The p-VHCF electrode shows a high stability during 5000 cycles (32 days) at a low current density of 2 A g⁻¹ with a capacity retention of 80.4 % (Fig. R3). This phenomenon

may be attributed to the following reasons. First is the irreversible capacity loss at the low current density [Ref. *Angew. Chem. Int. Ed.* 2023, e202303563]. The irreversible part of the capacity loss could be caused by the formation of a solid electrolyte interface on the electrode surfaces, where the passivation mechanism is more complicated in concentrated sulfuric acid. [Ref. *J. Mater. Chem. A*, 2020, 8, 21103–21109; *Electrochim. Acta.* 1993, 38, 981-987]. Secondly, the dissolution of active materials may cause the capacity fading. For instance, MnO₂ generally suffers from capacity loss due to the dissolution of Mn²⁺ from Mn³⁺ disproportionation, which is widely accepted [Refs. *Nat. Commun.* 2017, 8, 405; *Chem. Mater.* 2015, 27, 3609–3620; *Nat. Energy* 2016, 1, 16039]. In this work, it takes longer time of 32 days to performance the 5000 cycles at 2 A g⁻¹ compared with the cycle performance at 10 A g⁻¹ with 5 days, which may cause some dissolution of active materials. In addition, the dissolved oxygen may chemically oxidize the electrode materials in such a long-time test [Ref. *J. Mater. Chem. A*, 2019, 7, 20519–20539].

Fig. R3 Cycle performance at 2 A g⁻¹ for 5000 cycles (32 days) in 6 M H₂SO₄.

Comment 5: The abbreviation "H-VHCF" for pre-protonated vanadium hexacyanoferrate may be confusing for readers and does not clearly convey information about the material. The authors should select a more concise and transparent abbreviation.

Response: Many thanks for your suggestion. The HCF is the abbreviation "hexacyanoferrate", which is generally used in related literatures for the Prussian blue analogs [Ref. *Nat. Commun.* 2011, 2,550; *Chem. Rev.* 2021, 121, 11, 6654–6695; *Adv. Mater.* 2022, 34, 2105611]. "VHCF" means vanadium ion doped HCF. We agree with the reviewer that the "H-" can be misunderstanding. Therefore, to clearly convey information about the pre-protonated vanadium hexacyanoferrate, the abbreviation "p-VHCF" has been used in place of "H-VHCF" in the revised manuscript.

Comment 6: The manuscript's title is somewhat inaccurate and could be misleading. The electrochemical tests in the manuscript encompass two distinct parts: decoupling water splitting using H-VHCF as a redox mediator and coupling two half-reactions for electricity generation. Therefore, the statement "Decoupled Water/Hydrazine Electrolysis" is inappropriate and should be revised to accurately reflect the content.

Response: Thanks for your suggestion. Due to the limit of 15 words, it is kind of hard to clearly comprise the two parts in the title: decoupled water splitting using H-VHCF as a redox mediator as well as coupling two half-reactions for both hydrogen production and electricity generation. Therefore, we would like to emphasize one major innovation point, and the title has been revised as “**Decoupled electrolysis for hydrogen production and hydrazine oxidation via high-capacity and stable pre-protonated vanadium hexacyanoferrate**”.

Comment 7: Additionally, the manuscript should include a schematic representation of the coupling strategy for the reduction process of H-VHCF and hydrazine oxidation in the Scheme 1 to form a new electrolysis architecture, as this is a crucial aspect of the research that should be visually represented.

Response: According to your kind suggestion, a schematic representation including the coupling strategy for the reduction process of p-VHCF and hydrazine oxidation is added in the revised manuscript (**Fig. 1**).

[Revision]:

[Manuscript Page 5]:

Alternatively, the OER process can be replaced by the hydrazine oxidation reaction. As shown in Fig. 1b, the p-VHCF_{ox} formed during hydrogen production is coupled with the hydrazine oxidation process to generate a new electrolysis architecture to recycle the mediator. Considering the more positive chemical potential of the reduction of p-VHCF than the hydrazine oxidation reaction (chemical potential results are discussed below), the novel p-VHCF-N₂H₄ liquid battery can be formed, which enables hydrazine oxidation with simultaneous electricity generation (Step 2’).

Fig. 1 | Illustration of two-step decoupled electrolysis process. a Schematic of the hydrogen/oxygen production from decoupled acid water electrolysis with p-VHCF mediator electrode. **b** Schematic of decoupled electrolysis for hydrogen production and hydrazine oxidation with p-VHCF mediator electrode.

Response to the Reviewer #3

General comment: This work presents a decoupled acid water electrolysis system by a pre-protonated vanadium hexacyanoferrate (H-VHCF) solid-state redox mediator. Although the work is well done and the electrochemistry sound, I feel the work lacks novelty and does not constitute new science that is of interest to a broad readership targeted by Nature Communications. I think this should be published in a more specialist journal.

Response:

We thank the reviewer for the time and effort to review our manuscript. We have responded to the comments point by point and revised the manuscript accordingly. To clearly emphasize the novelty and new science, we have added more experimental results and discussion. We believe that the importance and novelty of the current work can be summarized into two aspects.

The first point is the design of a high-capacity and stable vanadium hexacyanoferrate (p-VHCF) mediator for decoupled water electrolysis, where the pre-protonation process expanded the hydrogen bond network for fast Grotthuss proton conduction. Thus, the electrode delivers a high reversible specific capacity up to 128 mAh g⁻¹ and excellent cycling performance of 6000 cycles with capacity retention of 100% at a current density of 10 A g⁻¹. Further, we add the rate performances at much higher current densities in the revised manuscript. The mediator shows an impressive capacity of 47 mAh g⁻¹ even at a high current density of 200 A g⁻¹. In addition, we tested the cycle performance of the p-VHCF electrode in various concentrations of H₂SO₄ electrolyte, and the p-VHCF electrode still exhibits better rate performance in 0.5 M H₂SO₄ electrolyte compared to reported CuFe TBA. Also, this material shows excellent cycling stability and a reservable redox capacity of 98.8 mAh g⁻¹ in H₃PO₄ aqueous solution, indicating a promising application at ultralow temperatures. Owing to the good performances in different conditions, the p-VHCF electrode shows efficient decoupled water electrolysis for hydrogen production.

The second innovation point is the development of a novel decoupled electrolysis system for hydrogen production and hydrazine oxidization, which realizes separate H₂ generation, electrical energy storage, and green treatment of hydrazine hazards. Typically, solar energy can be used to drive the Step 1 process for high-rate H₂ production at day-time. The hydrazine oxidation with electricity generation (Step 2') can be achieved through the p-VHCF-N₂H₄ liquid battery at night-time. In such a manner, the flexible energy conversion and storage using renewables can be expected. Meanwhile, the decoupled hydrogen production and hydrazine oxidation can operate at different rates, which is beneficial to meeting the requirements of varied applications. We have added more experimental results (by combining with Si solar cell) to demonstrate the feasibility of this hybrid energy conversion/storage concept. Therefore, we believe our work can receive broad interest from the

researchers working in different fields, such as material science, chemistry, renewable energy, *etc.*

Comment 1: This decoupled architecture was performed in 6 M H₂SO₄ electrolyte. For conventional acid water electrolysis, the electrolyte concentration is much lower than 6 M H₂SO₄, such as 0.5 M H₂SO₄, 1 M H₂SO₄, 0.1 M HClO₄. The high concentration of the electrolyte will put greater demands for electrolyzers and electrodes.

Response: As pointed out by the reviewer, the high concentration of the electrolyte in this work will put higher demands on electrolyzers and electrodes. Therefore, we further tested the cycle performance of the p-VHCF electrode in various concentrations of H₂SO₄ electrolyte, including 0.5, 1, 3, 5, and 6 M, at 10 A g⁻¹. As shown in **Supplementary Fig. 8**, the p-VHCF electrode delivers a specific capacity of 108 mAh g⁻¹ and 90% capacity retention at 3 M concentration after 6000 cycles. It also has a specific capacity of 95 mAh g⁻¹ and can maintain more than 75% capacity retention at 10 A g⁻¹ after 6000 cycles as the H₂SO₄ concentration decreases to 0.5 M. Those values remain favorable among the most currently reported mediator electrodes [*Refs. Adv. Energy Mater.* 2022, 2203455; *Energy Environ. Sci.*, 2021, 14, 4740–4759]. Generally, the properties of the current mediator are improved with the increased H₂SO₄ concentration. For the practical application, the reaction conditions should be comprehensively considered in terms of the balance of the cost and performance. Higher electrolyte concentration can be acceptable, and a good example is commercial alkaline water electrolysis, where 30 wt.% (around 6 M) KOH aqueous solution is used as the electrolyte [*Refs. J. Am. Chem. Soc.* 2023, 145, 21419–21431; *Nat. Commun.* 2022, 13, 7956].

In addition, we tested the performances of the p-VHCF electrode in 0.1 M HClO₄, 10 M H₃PO₄, and 2 M H₃PO₄ electrolytes. The p-VHCF electrode is unstable in 0.1 M HClO₄ (**Fig. R4**). However, as shown in **Supplementary Fig. 10a**, this material shows excellent cycling stability and a reservable redox capacity of 98.8 mAh g⁻¹ in 10 M H₃PO₄. It is reported that the freezing point of H₃PO₄ electrolyte can be lower than -80 °C with the increase of electrolyte concentration to 9 M. This result indicates that the p-VHCF mediator has a potential application at ultralow temperatures for H₂ production [*Refs. J. Am. Chem. Soc.* 2021, 143, 20302–20308; *Adv. Energy Mater.* 2020, 10, 2000968]. Moreover, this electrode is also stable in 2 M H₃PO₄ electrolyte **Supplementary Fig. 10b**.

Fig. R4 Cycle performance at 10 A g^{-1} for 6000 cycles in 0.1 M HClO_4 .

[Revision]:

[Manuscript Page 10-11]:

We further tested the cycle performance of the p-VHCF electrode in various concentrations of H_2SO_4 , including 0.5, 1, 3, 5, and 6 M, at 10 A g^{-1} . As shown in Supplementary Fig. 8, the p-VHCF electrode delivers a specific capacity of 108 mAh g^{-1} and 90% capacity retention at 3 M concentration after 6000 cycles. It also has a specific capacity of 95 mAh g^{-1} and can maintain more than 75% capacity retention at 10 A g^{-1} after 6000 cycles as the H_2SO_4 concentration decreases to 0.5 M. The rate performance of p-VHCF electrode at varied current densities in 0.5 M H_2SO_4 electrolyte was exhibited in Supplementary Fig. 9, and those values remain favorable among the most currently reported mediator electrode^{8,9}. Generally, the properties of the current mediator are improved with the increased H_2SO_4 concentration. For the practical application, the reaction condition should be comprehensively considered in terms of the balance of the cost and performance. Higher electrolyte concentration can be acceptable, and a good example is commercial alkaline water electrolysis, where 30 wt.% (around 6 M) KOH aqueous solution is used as the electrolyte^{42,43}. In addition, we tested the performances of the p-VHCF electrode in 10 M and 2 M H_3PO_4 electrolytes. As shown in Supplementary Fig. 10a, this material shows excellent cycling stability and a redox capacity of 98.8 mAh g^{-1} at 10 A g^{-1} in 10 M H_3PO_4 . It is reported that the freezing point of H_3PO_4 electrolyte can be lower than $-80 \text{ }^\circ\text{C}$ with the increase of electrolyte concentration to 9 M. This result indicates that the p-VHCF mediator has a potential application at ultralow temperatures for H_2 production^{34,44}. Moreover, this electrode is also stable in 2 M H_3PO_4 electrolyte (Supplementary Fig. 10b). Overall, owing to the good performances in different electrolytes, the p-VHCF electrode seems promising in decoupled electrolysis, batteries, and other electrochemical applications.

[SI Page S5, Supplementary Fig. 8]:

Supplementary Fig. 8 Cycle performance of the p-VHCF electrode at 10 A g⁻¹ for 6000 cycles in 0.5, 1, 3, 5 and 6 M H₂SO₄.

[SI Page S6, Supplementary Fig. 10]:

Supplementary Fig. 10 Cycling performance of the p-VHCF electrode in (a) 10 M and (b) 2 M H₃PO₄ electrolyte at 10 A g⁻¹.

Comment 2: There is no detailed introduction about why the authors using the hydrazine oxidation reaction (HzOR) to replace OER, the application of HzOR.

Response: Thanks for the comment. We have added more details about why hydrazine oxidation reaction (HzOR) was used to replace OER and the application of HzOR in the liquid battery in the introduction section.

[Revision]:

[Manuscript Page 3-4]:

Considering the sluggish kinetics of OER, the anodic OER can be replaced by other anodic oxidation reactions that are kinetically and thermodynamically favorable for energy-saving H₂ production, such as the methanol oxidation reaction¹⁹⁻²¹, the formate oxidation reaction^{22,23}, and the isopropanol oxidation reaction²⁴⁻²⁶. Among available options, hydrazine, as a liquid proton carrier, can be oxidized to inert nitrogen and release protons at the low voltage, effectively decreasing the input electrical energy^{27,28}. Besides, hydrazine oxidation reaction (HzOR) can combine with a reduction reaction with more positive chemical potential to form a hydrazine battery/cell for simultaneous generation of electricity²⁹⁻³¹. Inspired by this principle, combining hydrazine oxidation with the reduction of mediator electrode into a decoupled electrolysis system can enable hydrogen production and electricity generation, which may offer the possibility for the flexible energy conversion and storage using renewables. However, this topic in decoupled system is seldom investigated up to present.

Comment 3: The redox mediator and the decoupled process of this work is similar to the reported work of *Adv. Energy Mater.* 2021, 11, 2102057. Based on the Grotthuss proton conduction, the rate performance of H-VHCF is inferior to that of CuFe-TBA.

Response: Thanks for the comment. After careful comparison of our results with those reported in the literature [*Ref. Adv. Energy Mater.* 2021, 11, 2102057], it is found that our p-VHCF electrode exhibits better rate performance either in 6 M or 0.5 M H₂SO₄ electrolyte. Typically, we have retested the rate performance under the same current density conditions from 2 A g⁻¹ to 120 A g⁻¹ in 6 M and 0.5 M H₂SO₄. As shown in **Fig. R5a, b**, the p-VHCF electrode delivered reversible capacities of 152.7, 118.3, 102.2, 87.7 and 69.3 mAh g⁻¹ in 6 M H₂SO₄ and capacities of 111.6, 88.3, 77.5, 63.1 and 50 mAh g⁻¹ in 0.5 M H₂SO₄ at current densities of 2, 15, 40, 80 and 120 A g⁻¹, respectively. Those values are better compared with the rate performances of the CuFe TBA in **Fig. R5c**. Typically, the p-VHCF mediator shows an impressive capacity of 47 mAh g⁻¹ even at a high current density of 200 A g⁻¹ in 6 M H₂SO₄ electrolyte. The detailed rate performance values were summarized in **Table R1**. It is believed that the enhanced Grotthuss proton conduction due to the pre-protonated treatment contributed to the excellent performance.

Fig. R5 The rate performance of the p-VHCF electrode in (a) 6 M H₂SO₄ and (b) 0.5 M H₂SO₄. (c) The rate performance of CuFe-TBA electrode in 0.5 M H₂SO₄ reported in the reference [Ref. *Adv. Energy Mater.* 2021, 11, 2102057].

Table R1 Comparison of the electrode capacities at different current densities.

	2A g ⁻¹	15A g ⁻¹	40A g ⁻¹	80A g ⁻¹	120A g ⁻¹
p-VHCF in 6 M H ₂ SO ₄	152.7 mAh g ⁻¹	118.3 mAh g ⁻¹	102.2 mAh g ⁻¹	87.7 mAh g ⁻¹	69.3 mAh g ⁻¹
p-VHCF in 0.5 M H ₂ SO ₄	111.6 mAh g ⁻¹	88.3 mAh g ⁻¹	77.5 mAh g ⁻¹	63.1 mAh g ⁻¹	50 mAh g ⁻¹
CuFe TBA in 0.5 M H ₂ SO ₄	75.2 mAh g ⁻¹	69.6 mAh g ⁻¹	62.3 mAh g ⁻¹	53.6 mAh g ⁻¹	42.7 mAh g ⁻¹

[Revision]:

[Manuscript Page 9, Fig. 3c]

Even at a high current density of 200 A g⁻¹, a respectable capacity of 47 mAh g⁻¹ was still achieved, indicating ultrafast proton insertion/de-insertion rate. When the current density was up to 300 A g⁻¹,

the electrode only delivers a capacity of 17 mAh g⁻¹, indicating the limit of this electrode.

[Manuscript Page 11, Fig. 3c]

Fig. 3 | c Rate performance of the p-VHCF electrode for selected current densities.

[Manuscript Page 10]:

The rate performance of p-VHCF electrode at varied current density in 0.5 M H₂SO₄ electrolyte was exhibited in Supplementary Fig. 9, and those values remain favorable among the most currently reported mediator electrode^{8,9}.

[SI Page S6, Supplementary Fig. 9]

Supplementary Fig. 9 Rate performance of the p-VHCF electrode at varied current density in 0.5 M H₂SO₄ electrolyte.

Comment 4: Please provide the structure and morphology change, such as the XRD, XPS, and SEM images, after long cycling.

Response: According to your suggestion, the XRD, XPS, and SEM results of the p-VHCF mediator after 6000 cycles are given in **Supplementary Fig. 7**. It is found that there are not apparent changes of the structure and morphology after the long-term test, indicating the good stability of the mediator.

[Revision]:

[Manuscript Page 9-10]:

The XRD patterns, SEM images, and X-ray Photoelectron Spectroscopy (XPS) results after 6000 cycles were collected (Supplementary Fig. 7). There are not apparent changes of the structure and morphology after the long-term test, indicating the excellent stability of the p-VHCF electrode.

[SI Page S5, Supplementary Fig. 7]:

Supplementary Fig. 7 Structure and morphology of the p-VHCF electrode before and after the 6000 cycles. (a) XRD patterns. (b, c) SEM images before and after long cycling. (d) XPS survey spectra. (e, f) High-resolution XPS spectra of V 2p and Fe 2p.

Comment 5: From Figure 5b, the oxidation potential of HzOR is close to the reduction potential of HER, in practical it will consume low energy to produce hydrogen directly by hydrazine electrolysis.

Response:

Thanks for the comment. According to the Chronopotentiometry curve in **Fig. 6c**, our decoupled system enables to produce H_2 at a high rate (100 mA) with a **cell voltage input** of 1.05 V, followed by the hydrazine oxidation with an **output voltage** of 0.5 V. We also tested the voltages needed to produce hydrogen directly by hydrazine electrolysis at 100 mA. As shown in **Fig. R6**, 1.26 V is required for the direct hydrazine electrolysis with the proton exchange membrane. The large voltage can be attributed to the high electrochemical overpotentials of electrocatalysts, as well as the ohmic resistance of the solution and membrane.

Most importantly, we propose this decoupled system to enable the flexible energy conversion and storage. We can use solar energy to drive the Step 1 process for high-rate H_2 production at day-time, and achieve hydrazine oxidation with electricity generation (Step 2') through the novel p-VHCF- N_2H_4 liquid battery at night-time. Meanwhile, the decoupled hydrogen production and hydrazine oxidation can operate at different rates, which is beneficial to the requirements of varied applications.

To clarify this point, we built a solar cell driven decoupled electrolysis system and tested the performances of these two separate processes. As shown in **Fig. 6g**, the Step 1 for hydrogen production is driven by Si solar cell followed by the Step 2' for hydrazine oxidation with electricity generation. The operating current of solar cell driven Step 1 is 22.47 mA, which matches with the value estimated from the intersection of the LSV curve of the Step 1 for H_2 production and the I-V curve of the Si solar cell. Meanwhile, the Step 2' can output stable electricity (**Fig. 6h-i** and **Supplementary Fig. 22**). In this regard, we believe this novel design shows predictable potential for flexible energy storage and conversion (**Supplementary Fig. 23**).

Fig. R6 Voltage required for direct hydrazine electrolysis at 100 mA with a proton exchange membrane. [Cell structure: Pt coated Ti-mesh electrode | 0.5 M H_2SO_4 + 0.1 M N_2H_4 | Pt coated Ti-mesh electrode]

[Revision]:

[Manuscript Page 20, Fig. 6g-i]:

Fig. 6 | g Schematic of decoupled electrolysis system driven by solar cell in Step 1 for high-rate H₂ production at day-time, followed by the hydrazine oxidation with electricity generation in Step 2' at night-time. h The I-t curve of the Step 1 driven by Si solar cell. i The V-t curve of the Step 2' with a discharge current of 1 mA.

[Manuscript Page 21]:

More importantly, this decoupled system is proposed to enable the flexible energy conversion and storage. We can use solar energy to drive the Step 1 process for high-rate H₂ production at day-time, and achieve hydrazine oxidation with electricity generation (Step 2') through the novel p-VHCF-N₂H₄ liquid battery at night-time (Fig. 6g). To clarify this point, we built a Si solar cell driven decoupled electrolysis system and tested the performances of these two separate processes. As shown in Fig. 6h, the operating current of the solar cell driven Step 1 is 22.47 mA, which matches with the value estimated from the intersection of the LSV curve of the Step 1 for H₂ production and the current-voltage curve of the Si solar cell (Supplementary Fig. 22). Meanwhile, the Step 2' can output stable electricity (Fig. 6i). In this regard, we believe this novel design shows predictable potential for flexible energy conversion and storage compared with the direct hydrazine electrolysis (Supplementary Fig. 23).

[SI Page S12, Supplementary Fig. 22]:

Supplementary Fig. 22 LSV curve of the Step 1 for H₂ production and the I-V curve of the Si solar cell under simulated AM 1.5 G light illumination.

[SI Page S13, Supplementary Fig. 23]:

Supplementary Fig. 23 Schematic of flexible energy conversion and storage system by applying decoupled electrolysis.

Comment 6: For the decoupled hydrazine electrolysis process, the authors indicate that this process could remove hydrazine from acid waste water. Owing to the limited capacity of H-VHCF, how long will it take to remove hydrazine to a safe concentration? In addition, it is meaningless to employ this process as a secondary battery due to its lower discharge voltage.

Response: Thanks for the comment. We used the *Watt and Chrisp* method [Ref. *Anal. Chem.* 1995, 24 (12), 2006–2008] to determine the concentration of hydrazine specie. Typically, 2 mL hydrazine solution was mixed with 0.134 M 4-Dimethylaminobenzaldehyde ethanol solution (2 mL), and stored for 20 min for UV-vis test. As shown in **Fig. R7**, the peak located at 458 nm is the absorption peak of p-Dimethylaminobenzodiazine from the reaction of hydrazine and 4-Dimethylaminobenzaldehyde. 100 ppm hydrazine in 100 mL of 0.5 M H₂SO₄ electrolyte was used as the initial solution to simulate

the industrial hydrazine sewage. The chronoamperometry test of Step 2' was performed at a current of 20 mA. After 7 h, the peak disappeared means there is no detectable hydrazine species in the reaction solution. Note that the higher concentration of hydrazine was used in the decoupled electrolysis process for stable electricity generation via the p-VHCF-N₂H₄ liquid battery, which needs longer time for hydrazine oxidation. For decoupled systems, the capacity of the mediator can be flexibly adjusted by changing the mass loading of the mediator electrode according to the literatures [Refs. *Angew. Chem. Int. Ed.* 2018, 57, 2904–2908; *Angew. Chem. Int. Ed.* 2019, 58, 4622–4626; *Angew. Chem. Int. Ed.* 2023, e202303563]. For the practical application, we can increase the loading amount of the mediator material on one single electrode and/or use multiple mediator electrodes for the continuous hydrazine oxidation. In this work, we first propose this idea, which definitely needs more engineering optimization in our next study.

Fig. R7 The UV-vis absorption spectra of (a) the hydrazine standard solution (the insert is the calibration curve), and (b) the experimental curves obtained at different hydrazine treatment time using 100 mL of acid electrolyte with 100 ppm hydrazine as the reaction electrolyte. Hydrazine standard solution with different concentrations was obtained by dissolving different amount of hydrazine in 0.5 M H₂SO₄. Typically, the hydrazine can react with 4-Dimethylaminobenzaldehyde to generate p-Dimethylaminobenzodiazine, which shows yellow color and can be detected by UV-vis test.

In addition, regarding the lower voltage in Step 2' that pointed out by the reviewer, the voltage of the liquid battery in the manuscript was obtained after Step 1 (H₂ production) at 100 mA with a charge capacity of 2.78 mAh, where the p-VHCF mediator electrode was charged to 1.0 V (vs. Ag/AgCl). When the p-VHCF electrode was charged to 1.2 V (vs. Ag/AgCl), the battery can deliver an open-circuit voltage of 1.05 V (**Supplementary Fig. 21**).

Meanwhile, considering the water splitting potential, the aqueous batteries, especially aqueous proton batteries with only the proton charge carriers usually hold an electrochemical window of 0-1.2

V, but those secondary batteries intrinsically own the perceived merits of high safety, low cost, easy manufacture, fast kinetics, and long-term cycling stability, and deserve more studies [Refs. *Adv. Mater.* 2022, 34, 2207747; *ACS Nano.* 2023, 17, 10965-10975; *Angew. Chem. Int. Ed.* 2023, 62, e202300390; *Nat. Commun.* 2020, 11, 959; *Energy Storage Mater.* 2021, 36, 1-9; *J. Am. Chem. Soc.* 2021, 143, 20302-20308; *Nano Lett.* 2023, 23, 9664-9671].

Therefore, we believe this novel battery system formed in Step 2' is suitable as a rechargeable battery through the delicate control of the series-parallel connection. Even for commercial PEM fuel cell, the output voltage of one single cell is generally around 0.6~0.7 V, and the series connection is needed to get higher voltages for the practical application.

In addition, **Table R2** presented the open-circuit voltages of the recent reported aqueous batteries, which can further illustrate our points.

[Revision]:

[Manuscript Page 19]:

The formed p-VHCF-N₂H₄ battery can deliver an open-circuit voltage about 0.84 V. When the p-VHCF electrode is charged to 1.2 V (vs. Ag/AgCl), the battery can deliver an open-circuit voltage of 1.05 V (Supplementary Fig. 21). The aqueous batteries, especially aqueous proton batteries with only the proton charge carriers, usually hold an electrochemical window of 0-1.2 V, but those secondary batteries intrinsically own the perceived merits of high safety, low cost, easy manufacture, fast kinetics, and long-term cycling stability^{43-44,52-56}.

[SI Page S12, Supplementary Fig. 21]:

Supplementary Fig. 21 The open-circuit voltage of the p-VHCF-N₂H₄ battery in Step 2' after p-VHCF electrode was charged to different potentials.

Table R2. The parameters of the recently reported aqueous batteries.

Cathode	Anode	Electrolyte	Open-circuit voltage	Ref.
p-VHCF	Pt	6 M H ₂ SO ₄ /0.5 M H ₂ SO ₄ + N ₂ H ₄	1.05 V (Charged to 1.2 V vs. Ag/AgCl)	This work
Mo ₂ C/Ni@C/CS	Zn	1 M KOH+0.02M Zn ²⁺ 1 M KOH+0.2 M N ₂ H ₄	0.366 V	1
NiCoP/NF	Zn foil	1.0 M KOH + 0.02 M Zn(CH ₃ COO) ₂ 1.0 M KOH + 0.2 M N ₂ H ₄	0.315 V	2
Bimetallic RuCo precatalyst	RuCo precatalyst	hydrazine-nitrate flow battery	0.94 V	3
MnO ₂ @GF	PTO	2 M MnSO ₄ + 2 M H ₂ SO ₄	0.94 V	4
pDTP-NQ	pDTP-AQ	1 M H ₂ SO ₄	0.9 V	5
CuFe-TBA	Pt/C catalyst	9 M H ₃ PO ₄	1.2 V	6
benzoquinone	H ₂	5 M H ₂ SO ₄	1 V	7

References

- Feng, Y., Shi, Q., Lin, J., Chai, E., Zhang, X., Liu, Z., Jiao, L. and Wang, Y. Decoupled Electrochemical Hydrazine “Splitting” via a Rechargeable Zn–Hydrazine Battery. *Adv. Mater.* **34**, 2207747 (2022).
- Wang, H.-Y., Wang, L., Ren, J.-T., Tian, W., Sun, M., Feng, Y. and Yuan, Z.-Y. Taking Advantage of Potential Coincidence Region: Advanced Self-Activated/Propelled Hydrazine-Assisted Alkaline Seawater Electrolysis and Zn–Hydrazine Battery. *ACS Nano.* **17**, 10965-10975 (2023).
- Zhu, W., Zhang, X., Yao, F., Huang, R., Chen, Y., Chen, C., Fei, J., Chen, Y., Wang, Z. and Liang, H. A Hydrazine-Nitrate Flow Battery Catalyzed by a Bimetallic RuCo Precatalyst for Wastewater Purification along with Simultaneous Generation of Ammonia and Electricity. *Angew. Chem. Int. Ed.* **62**, e202300390 (2023).
- Guo, Z., Huang, J., Dong, X., Xia, Y., Yan, L., Wang, Z. and Wang, Y. An Organic/Inorganic Electrode-based Hydronium-ion Battery. *Nat. Commun.* **11**, 959 (2020).
- Wang, X., Zhou, J. and Tang, W. Poly(dithieno[3,2-b:2',3'-d]pyrrole) Twisting Redox Pendants

Enabling High Current Durability in All-organic Proton Battery. *Energy Storage Mater.* **36**, 1-9 (2021).

6. Zhu, Z., Wang, W., Yin, Y., Meng, Y., Liu, Z., Jiang, T., Peng, Q., Sun, J. and Chen, W. An Ultrafast and Ultra-Low-Temperature Hydrogen Gas-Proton Battery. *J. Am. Chem. Soc.* **143**, 20302-20308 (2021).
7. Liu, S., Jin, S., Jiang, T., Sajid, M., Xu, J., Zhang, K., Fan, Y., Peng, Q., Zheng, X., Xie, Z., Liu, Z., Zhu, Z., Wang, X., Nian, Q., Chen, J., Li, K., Shen, C. and Chen, W. Aqueous Organic Hydrogen Gas Proton Batteries with Ultrahigh-Rate and Ultralow-Temperature Performance. *Nano Lett.* **23**, 9664-9671 (2023).

Comment 7: In page 11 and 12, the unit of the electrode area should be cm².

Response: Thanks for the comment. We have corrected this error and made a careful check.

Comment 8: The format of references should be unified.

Response: Thanks for your kind suggestions. We have unified the format of all references.

REVIEWERS' COMMENTS

Reviewer #1 (Remarks to the Author):

This reviewer acknowledges the receipt of the revised manuscript. The authors conducted substantial experiments to supplement the evidence. In situ Raman spectroscopy results are interesting and provide redox information of the investigated materials. A potential window to see the stability range of this material supports strongly its high stability. This material as reported is among the most efficient redox mediators for water splitting. It is recommended for publication.

Reviewer #2 (Remarks to the Author):

I am satisfied with the revision and the paper now can be recommended for publication.

Reviewer #3 (Remarks to the Author):

The authors have adequately responded to my concerns and I recommend this paper for publication.